# DIPPER, a spatiotemporal proteomics atlas of human intervertebral discs for exploring ageing and degeneration dynamics

Vivian Tam[1,2†], Peikai Chen[1†‡], Anita Yee[1], Nestor Solis[3], Theo Klein[3§], Mateusz Kudelko[1], Rakesh Sharma[4], Wilson CW Chan[1,2,5], Christopher M Overall[3], Lisbet Haglund[6], Pak C Sham[7], Kathryn Song Eng Cheah[1], Danny Chan[1,2*]

[1]School of Biomedical Sciences, , The University of Hong Kong, Hong Kong; [2]The University of Hong Kong Shenzhen of Research Institute and Innovation (HKU-SIRI), Shenzhen, China; [3]Centre for Blood Research, Faculty of Dentistry, University of British Columbia, Vancouver, Canada; [4]Proteomics and Metabolomics Core Facility, The University of Hong Kong, Hong Kong; [5]Department of Orthopaedics Surgery and Traumatology, HKU-Shenzhen Hospital, Shenzhen, China; [6]Department of Surgery, McGill University, Montreal, Canada; [7]Centre for PanorOmic Sciences (CPOS), The University of Hong Kong, Hong Kong

**\*For correspondence:**
chand@hku.hk

[†]These authors contributed equally to this work

**Present address:** [‡]Department of Orthopaedics Surgery and Traumatology, HKU-Shenzhen Hospital, Shenzhen, China; [§]Triskelion BV, Zeist, Netherlands

**Abstract** The spatiotemporal proteome of the intervertebral disc (IVD) underpins its integrity and function. We present DIPPER, a deep and comprehensive IVD proteomic resource comprising 94 genome-wide profiles from 17 individuals. To begin with, protein modules defining key directional trends spanning the lateral and anteroposterior axes were derived from high-resolution spatial proteomes of intact young cadaveric lumbar IVDs. They revealed novel region-specific profiles of regulatory activities and displayed potential paths of deconstruction in the level- and location-matched aged cadaveric discs. Machine learning methods predicted a 'hydration matrisome' that connects extracellular matrix with MRI intensity. Importantly, the static proteome used as point-references can be integrated with dynamic proteome (SILAC/degradome) and transcriptome data from multiple clinical samples, enhancing robustness and clinical relevance. The data, findings, and methodology, available on a web interface (http://www.sbms.hku.hk/dclab/DIPPER/), will be valuable references in the field of IVD biology and proteomic analytics.

## Introduction

The 23 intervertebral discs (IVDs) in the human spine provide stability, mobility, and flexibility. IVD degeneration (IDD), most common in the lumbar region (*Saleem et al., 2013*; *Teraguchi et al., 2014*), is associated with a decline in function and a major cause of back pain, affecting up to 80% of the world's population at some point in life (*Rubin, 2007*), presenting significant socioeconomic burdens. Multiple interacting factors such as genetics, influenced by ageing, mechanical and other stress factors, contribute to the pathobiology, onset, severity, and progression of IDD (*Munir et al., 2018*).

IVDs are large, avascular, extracellular matrix (ECM)-rich structures comprising three compartments: a hydrated nucleus pulposus (NP) at the centre, surrounded by a tough annulus fibrosus (AF) at the periphery, and cartilaginous endplates of the adjoining vertebral bodies (*Humzah and Soames, 1988*). The early adolescent NP is populated with vacuolated notochordal-like cells, which are gradually replaced by small chondrocyte-like cells (*Risbud et al., 2015*). Blood vessels terminate

**eLife digest** The backbone of vertebrate animals consists of a series of bones called vertebrae that are joined together by disc-like structures that allow the back to move and distribute forces to protect it during daily activities. It is common for these intervertebral discs to degenerate with age, resulting in back pain and severely reducing quality of life.

The mechanical features of intervertebral discs are the result of their proteins. These include extracellular matrix proteins, which form the external scaffolding that binds cells together in a tissue, and signaling proteins, which allow cells to communicate. However, how the levels of different proteins in each region of the disc vary with time has not been fully examined.

To establish how protein composition changes with age, Tam, Chen et al. quantified the protein levels and gene activity (which leads to protein production) of intervertebral discs from young and old deceased individuals. They found that the position of different mixtures of proteins in the intervertebral disc changes with age, and that young people have high levels of extracellular matrix proteins and signaling proteins. Levels of these proteins decreased as people got older, as did the amount of proteins produced.

To determine which region of the intervertebral disc different proteins were in, Tam, Chen et al. also performed magnetic resonance imaging (MRI) of the samples to correlate image intensity (which represents water content) with the corresponding protein signature. The data obtained provides a high-quality map of how the location of different proteins changes with age, and is available online under the name DIPPER. This database is an informative resource for research into skeletal biology, and it will likely advance the understanding of intervertebral disc degeneration in humans and animals, potentially leading to the development of new treatment strategies for this condition.

at the endplates, nourishing and oxygenating the NP via diffusion, whose limited capacity mean that NP cells are constantly subject to hypoxic, metabolic, and mechanical stresses (*Urban et al., 2004*).

With ageing and degeneration, there is an overall decline in cell 'health' and numbers (*Rodriguez et al., 2011*; *Sakai et al., 2012*), disrupting homeostasis of the disc proteome. The ECM has key roles in biomechanical function and disc hydration. Indeed, a hallmark of IDD is reduced hydration in the NP, diminishing the disc's capacity to dissipate mechanical loads. Clinically, T-2 weighted MRI is the gold standard for assessing IDD, that uses disc hydration and structural features such as bulging or annular tears to measure severity (*Pfirrmann et al., 2001*; *Schneiderman et al., 1987*). The hydration and mechanical properties of the IVD are dictated by the ECM composition, which is produced and maintained by the IVD cells.

To meet specific biomechanical needs, cells in the IVD compartments synthesise different compositions of ECM proteins. Defined as the 'matrisome' (*Naba et al., 2012*), the ECM houses the cells and facilitates their inter-communication by regulation of the availability and presentation of signalling molecules (*Taha and Naba, 2019*). With ageing or degeneration, the NP becomes more fibrotic and less compliant (*Yee et al., 2016*), ultimately affecting disc biomechanics (*Newell et al., 2017*). Changes in matrix stiffness can have a profound impact on cell-matrix interactions and downstream transcriptional regulation, signalling activity, and cell fate (*Park et al., 2011*).

The resulting alterations in the matrisome lead to vicious feedback cycles that reinforce cellular degeneration and ECM changes. Notably, many of the associated IDD genetic risk factors, such as COL9A1 (*Jim et al., 2005*), ASPN (*Song et al., 2008*), and CHST3 (*Song et al., 2013*), are variants in genes encoding matrisome proteins, highlighting their importance for disc function. Therefore, knowledge of the cellular and extracellular proteome and their spatial distribution in the IVD is crucial to understanding the mechanisms underlying the onset and progression of IDD (*Feng et al., 2006*).

Current knowledge of IVD biology is inferred from a limited number of transcriptomic studies on human (*Minogue et al., 2010*; *Riester et al., 2018*; *Rutges et al., 2010*) and animal (*Veras et al., 2020*) discs. Studies showed that cells in young healthy NP express markers including CD24, KRT8, KRT19, and T (*Fujita et al., 2005*; *Minogue et al., 2010*; *Rutges et al., 2010*), whereas NP cells in aged or degenerated discs have different and variable molecular signatures (*Chen et al., 2006*;

*Rodrigues-Pinto et al., 2016*), such as genes involved in TGFβ signalling (TGFA, INHA, INHBA, BMP2/6). The healthy AF expresses genes including collagens (COL1A1 and COL12A1) (*van den Akker et al., 2017*), growth factors (PDGFB, FGF9, VEGFC), and signalling molecules (NOTCH and WNT) (*Riester et al., 2018*). Although transcriptomic data provides valuable cellular information, it does not faithfully reflect the molecular composition. Cells represent only a small fraction of the disc volume, transcriptome-proteome discordance does not enable accurate predictions of protein levels from mRNA (*Fortelny et al., 2017*), and the disc matrisome accumulates and remodels over time.

Proteomic studies on animal models of IDD, including murine (*McCann et al., 2015*), canine (*Erwin et al., 2015*), and bovine (*Caldeira et al., 2017*), have been reported. Nevertheless, human-animal differences in cellular phenotypes and mechanical loading physiologies mean that these findings might not translate to the human scenario. So far, human proteomic studies have compared IVDs with other cartilaginous tissues (*Önnerfjord et al., 2012*) and have shown increases in fibrotic changes in aging and degeneration (*Yee et al., 2016*), a role for inflammation in degenerated discs (*Rajasekaran et al., 2020*), the presence of haemoglobins and immunoglobulins in discs with spondylolisthesis and herniation (*Maseda et al., 2016*), and changes in proteins related to cell adhesion and migration in IDD (*Sarath Babu et al., 2016*). The reported human disc proteomes were limited in the numbers of proteins identified and finer compartmentalisation within the IVD, and disc levels along the lumbar spine have yet to be studied. Nor have the proteome dynamics in term of ECM remodelling (synthesis and degradation) in young human IVDs and changes in ageing and degeneration been described.

In this study, we presented DIPPER (the Big Dipper are point-reference stars for guiding nautical voyages), a comprehensive disc proteomic resource, comprising static spatial proteome, dynamic proteome and transcriptome, and a methodological flow, for studying the human IVD in youth, ageing, and degeneration. First, we established a high-resolution point-reference map of static spatial proteomes along the lateral and anteroposterior directions of IVDs at three lumbar levels, contributed by a young (16M) and an aged (59M) cadavers with no reported scoliosis or degeneration. We evaluated variations among the disc compartments and levels by principal component analysis (PCA), analysis of variance (ANOVA), and identification of differentially expressed proteins (DEPs). We discovered modules containing specific sets of proteins that describe the directional trends of a young IVD, and the deconstruction of these modules with ageing and degeneration. Using a LASSO regression model, we identified proteins (the hydration matrisome) predictive of tissue hydration as indicated by high-resolution MRI of the aged discs. Finally, we showed how the point-reference proteomes can be utilised, to integrate with other independent transcriptome and dynamic proteome (SILAC and degradome) datasets from 15 additional clinical disc specimens, elevating the robustness of the proteomic findings. An explorable web interface hosting the data and findings is presented, serving as a useful resource for the scientific community.

## Results

### Disc samples and their phenotypes

DIPPER comprises 94 genome-wide measurements from lumbar disc components of 17 individuals (*Figure 1A*; *Table 1*), with data types ranging from label-free proteomics, transcriptomics, SILAC to degradome (*López-Otín and Overall, 2002*). High-resolution static spatial proteomes were generated from multiple intact disc segments of young trauma-induced (16 M) and aged (57 M) cadaveric spines. T1- and T2-weighted MRI (3T) showed the young discs (L3/4, L4/5, L5/S1) were non-degenerated, with a Schneiderman score of 1 from T2 images (*Figure 1B*). The NP of young IVD were well hydrated (white) with no disc bulging, endplate changes, or observable inter-level variations (*Figure 1B*; *Figure 1—figure supplement 1A*), consistent with healthy discs and were deemed fit to serve as a benchmarking point-reference. To investigate structural changes associated with ageing, high-resolution (7T) MRI was taken for the aged discs (*Figure 1C*). All discs had irregular endplates, and annular tears were present (green arrowheads) adjacent to the lower endplate and extending towards the posterior region at L3/4 and L4/5 (*Figure 1C*). The NP exhibited regional variations in hydration in both sagittal and transverse images (*Figure 1C*). Morphologically, the aged discs were less hydrated and the NP and AF structures less distinct, consistent with gross observations (*Figure 1—figure supplement 1A*). Scoliosis was not detected in these two individuals.

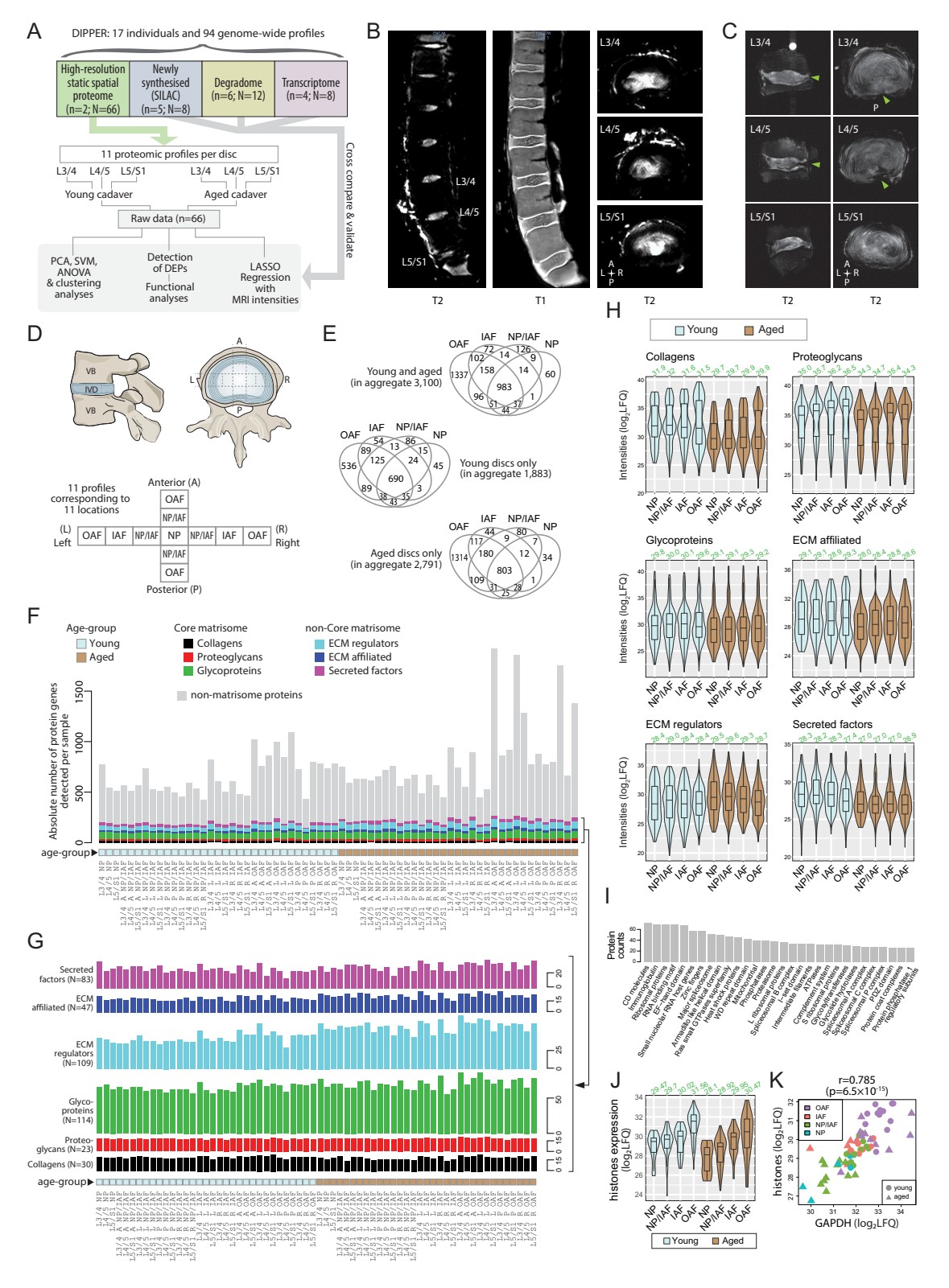

**Figure 1.** Outline of samples, workflow, MRI, and global overview of data in DIPPER. (**A**) Schematic diagram showing the structure of the samples, data types, and flow of analyses in DIPPER. *n* is the number of individuals. *N* is the number of genome-wide profiles. (**B**) Clinical T2-weighted MRI images (3T) of the young lumbar discs in the sagittal and transverse plane (left and right panels), T1 MRI image of the young lumbar spine (middle panel). (**C**) High-resolution (7T) T2-weighted MRI of the aged lower lumbar spine in sagittal (left panel) and transverse plane (right panel). (**D**) Diagram showing the

*Figure 1 continued on next page*

*Figure 1 continued*

anatomy of the IVD and locations where the samples were taken. VB: vertebral body; NP, nucleus pulposus; AF, annulus fibrosus; IAF: inner AF; OAF: outer AF; NP/IAF: a transition zone between NP and IAF. (**E**) Venn diagrams showing the overlaps of detected proteins in the four major compartments. Top panel, young and aged profiles; middle, young only; bottom, aged only. (**F**) Barchart showing the numbers of proteins detected per sample, categorised into matrisome (coloured) or non-matrisome proteins (grey). (**G**) Barcharts showing the composition of the matrisome and matrisome-associated proteins. Heights of bars indicate the number of proteins in each category expressed per sample. The N number in brackets indicate the aggregate number of proteins. (**H**) Violin plots showing the level of subcategories of ECMs in different compartments of the disc. The green number on top of each violin shows its median. LFQ: label-free quantification. (**I**) Top 30 HGNC gene families for all non-matrisome proteins detected in the dataset. (**J**) Violin plots showing the averaged expression levels of 10 detected histones across the disc compartments and age-groups. (**K**) Scatter-plot showing the co-linearity between GAPDH and histones.

The online version of this article includes the following figure supplement(s) for figure 1:

**Figure supplement 1.** Gross images of the discs and overview of the static spatial proteomic data.

**Figure supplement 2.** Numbers of proteins detected and protein levels for each combination of sample groups.

**Figure supplement 3.** Heatmaps showing different categories of detected proteins.

**Figure supplement 4.** Histones and housekeeping genes reflect cellularities.

Information of the disc samples used in the generation of other profiling data are described in Materials and methods and *Table 1*. They are clinical samples taken from patients undergoing surgery. The disc levels and intactness varied, and thus are more suitable for cross-validation purposes and some are directly relevant to IDD.

## Quality data detects large numbers of matrisome and non-matrisome proteins

The intact discs from the two cadaveric spines enabled us to derive spatial proteomes for young and aged human IVDs. We subdivided each lumbar disc into 11 key regions (*Figure 1D*), spanning the

**Table 1.** Summary of disc samples in DIPPER.

| Samples | Age | Sex | Disc level/s | Disc regions | Reason for surgery |
|---|---|---|---|---|---|
| *Cadaver samples* | | | | | |
| Young spine | 16 | M | L3/4, L4/5, L5/S1 | NP, NP/IAF, IAF, OAF | N/A |
| Aged spine | 59 | M | L3/4, L4/5, L5/S1 | NP, NP/IAF, IAF, OAF | N/A |
| *Transcriptome samples* | | | | | |
| YND74 | 17 | M | L1/2 | NP, OAF | Scoliosis |
| YND88 | 16 | M | L1/2 | NP, OAF | Scoliosis |
| AGD40 | 62 | F | L4/5 | NP, OAF | Degeneration |
| AGD45 | 47 | M | L4/5 | NP, OAF | Degeneration |
| *SILAC samples* | | | | | |
| YND148 | 19 | F | L2/3 | OAF | Scoliosis |
| YND149 | 15 | F | L1/2 | OAF | Scoliosis |
| YND151 | 15 | F | L1/2 | NP, OAF | Scoliosis |
| YND152 | 14 | F | L1/2 | NP, OAF | Scoliosis |
| AGD80 | 63 | M | L4/5 | NP, OAF | Degeneration |
| *Degradome samples* | | | | | |
| YND136 | 17 | F | L1/2 | NP, OAF | Scoliosis |
| YND141 | 20 | F | L1/2 | NP, OAF | Scoliosis |
| AGD143 | 53 | M | L1/2 | NP, OAF | Trauma |
| AGD62 | 55 | F | L5/S1 | NP, OAF | Degeneration |
| AGD65 | 68 | F | L4/5 | NP, OAF | Degeneration |
| AGD67 | 55 | M | L4/5 | NP, OAF | Degeneration |

outer-most (outer AF; OAF) to the central-most (NP) region of the disc, traversing both anteroposterior and lateral axes, adding valuable spatial information to our proteomic dataset. Since the disc is an oval shape, an inner AF (IAF) region was assigned in the lateral directions. A 'mixed' compartment between the NP and IAF with undefined boundary was designated as the NP/IAF in all four (anteroposterior and lateral) directions. In all, this amounted to 66 specimens with different compartments, ages, directions and levels, which then underwent LC-MS/MS profiling (*Supplementary file 1*). Systematic analyses of the 66 profiles are depicted in a flowchart (*Figure 1A*).

A median of 654 proteins per profile were identified for the young samples and 829 proteins for the aged samples, with a median of 742 proteins per profile for young and aged combined. The proteome-wide distributions were on similar scales across the profiles (*Figure 1—figure supplement 1B*). Of the 3100 proteins detected in total, 418 were matrisome proteins (40.7% of all known matrisome proteins) and 2682 non-matrisome proteins (~14% of genome-wide non-matrisome genes) (*Figure 1E*; *Figure 1—figure supplement 1C*), and 983 were common to all four major compartments, namely the OAF, IAF, NP/IAF, and NP (*Figure 1E*, upper panel). A total of 1883 proteins were identified in young discs, of which 690 (36%) were common to all regions. Additionally, 45 proteins (2.4%) were unique to NP, whilst NP/IAF, IAF, and OAF had 86 (4.6%), 54 (2.9%), and 536 (28%) unique proteins, respectively (*Figure 1E*, middle panel). For the aged discs, 2791 proteins were identified, of which 803 (28.8%) were common to all regions. NP, NP/IAF, and IAF had 34 (12%), 80 (28.7%), and 44 (15%) unique proteins, respectively, with the OAF accounting for the highest proportion of 1314 unique proteins (47%) (*Figure 1E*, lower panel). The aged OAF had the highest number of detected proteins with an average of 1156, followed by the young OAF with an average of 818 (*Figure 1F*). The quantity and spectrum of protein categories identified suggest sufficient proteins had been extracted and the data are of high quality.

## Levels of matrisome proteins decline in all compartments of aged discs

We divided the detected matrisome proteins into core matrisome (ECM proteins, encompassing collagens, proteoglycans, and glycoproteins) and non-core matrisome (ECM regulators, ECM affiliated, and secreted factors), according to a matrisome classification database (*Naba et al., 2012*) (matrisomeproject.mit.edu) (*Figure 1F*). Despite the large range of total numbers of proteins detected (419 to 1920) across the 66 profiles (*Figure 1F*), all six subcategories of the matrisome contained similar numbers of ECM proteins (*Figure 1F and G*; *Figure 1—figure supplement 2A*). The non-core matrisome proteins were significantly more abundant in aged than in young discs (*Figure 1—figure supplement 2A*). On average, 19 collagens, 18 proteoglycans, 68 glycoproteins, 52 ECM regulators, 22 ECM affiliated proteins, and 29 secreted factors of ECM were detected per profile. The majority of the proteins in these matrisome categories were detected in all disc compartments, and in both age groups. A summary of all the comparisons are presented in *Figure 1—figure supplement 1C–E*, and the commonly expressed matrisome proteins are listed in *Table 2*.

Even though there are approximately three times more non-matrisome than matrisome proteins per profile on average (*Figure 1F*), their expression levels in terms of label-free quantification (LFQ) values are markedly lower (*Figure 1—figure supplement 2B*). Specifically, the expression levels of core-matrisome were the highest, with an average $\log_2$(LFQ) of 30.65, followed by non-core matrisome at 28.56, and then non-matrisome at 27.28 (*Figure 1—figure supplement 2B*). Within the core-matrisome, the expression was higher (p=$6.4 \times 10^{-21}$) in young (median 30.74) than aged (median 29.72) discs (*Figure 1—figure supplement 2C & D*). This difference between young and aged discs is consistent within the subcategories of core and non-core matrisome, with the exception of the ECM regulator category (*Figure 1H*). The non-core matrisome and non-matrisome, however, exhibited smaller cross-compartment and cross-age differences in terms of expression levels (*Figure 1—figure supplement 2E–H*). That is, the levels of ECM proteins in each compartment of the disc declines with ageing and possibly changes in the relative composition, while the numbers of proteins detected per matrisome subcategory remain similar. This agrees with the concept that with ageing, ECM synthesis is not sufficient to counterbalance degradation, as exemplified in a proteoglycan study (*Silagi et al., 2018*).

**Table 2.** Commonly expressed extracellular matrix (ECM) and associated proteins across all 66 profiles in the spatial proteome.

| Categories (Number of proteins) | Protein names |
|---|---|
| *Core matrisome* | |
| Collagens (13) | COL1A1/2, COL2A1, COL3A1, COL5A1, COL6A1/2/3, COL11A1/2, COL12A1, COL14A1, COL15A1 |
| Proteoglycans (14) | ACAN, ASPN, BGN, CHAD, DCN, FMOD, HAPLN1, HSPG2, LUM, OGN, OMD, PRELP, PRG4, VCAN |
| Glycoproteins (34) | ABI3BP, AEBP1, CILP, CILP2, COMP, DPT, ECM2, EDIL3, EFEMP2, EMILIN1, FBN1, FGA, FGB, FN1, FNDC1, LTBP2, MATN2/3, MFGE8, MXRA5, NID2, PCOLCE, PCOLCE2, PXDN, SMOC1/2, SPARC, SRPX2, TGFBI, THBS1/2/4, TNC, TNXB |
| *Other matrisome* | |
| ECM affiliated proteins (10) | ANXA1/2/4/5/6, CLEC11A, CLEC3A/B, CSPG4, SEMA3C |
| ECM regulators (16) | A2M, CD109, CST3, F13A1, HTRA1, HTRA3, ITIH5, LOXL2/3, PLOD1, SERPINA1/3/5, SERPINE2, SERPING1, TIMP1 |
| Secreted factors (2) | ANGPTL2, FGFBP2 |

## Cellular activities inferred from non-matrisome proteins

Although 86.5% (2682) of the detected proteins were non-matrisome, their expression levels were considerably lower than matrisome proteins across all sample profiles (*Figure 1F*). A functional categorisation according to the Human Genome Nomenclature Committee gene family annotations (*Yates et al., 2017*) showed many categories containing information for cellular components and activities, with the top 30 listed in *Figure 1I*. These included transcriptional and translational machineries, post-translational modifications, mitochondrial function, protein turnover and importantly, transcriptional factors, cell surface markers, and inflammatory proteins that can inform gene regulation, cell identity, and response in the context of IVD homeostasis, ageing, and degeneration.

These functional overviews highlighted 77 DNA-binding proteins and/or transcription factors, 83 cell surface markers, and 175 inflammatory-related proteins, with their clustering data presented as heatmaps (*Figure 1—figure supplement 3*). Transcription factors and cell surface markers are detected in some profiles (*Figure 1—figure supplement 3A and B*). The heatmap of the inflammatory-related proteins showed that more than half of the proteins are detected in the majority of samples, with four major clusters distinguished by age and expression levels (*Figure 1—figure supplement 3C*). For example, one of the clusters in the aged samples showed enrichment for complement and coagulation cascades (False Discovery Rate, FDR q = $1.62 \times 10^{-21}$) and clotting factors (FDR q = $6.05 \times 10^{-9}$), indicating potential infiltration of blood vessels. Lastly, there are 371 proteins involved in signalling pathways, and their detection frequency in the different compartments and heat map expression levels are illustrated in *Figure 1—figure supplement 3D*.

## Histones and housekeeping genes inform compartment- and age-specific variations in cellularity

Cellularity within the IVD, especially the NP, decreases with age and degeneration (*Rodriguez et al., 2011*; *Sakai et al., 2012*). We assessed whether cellularity of the different compartments could be inferred from the proteomic data. Quantitation of histones can reflect the relative cellular content of tissues (*Wiśniewski et al., 2014*). We detected 10 histones, including subunits of histone 1 (HIST1H1B/C/D/E, HIST1H2BL, HIST1H3A, HIST1H4A) and histone 2 (HIST2H2AC, HIST2H2BE, HIST2H3A), with four subunits identified in over 60 sample profiles that are mutually co-expressed (*Figure 1—figure supplement 4A*). Interestingly, histone concentrations, and thus cellularity, increased from the inner to the outer compartments of the disc, and showed a highly significant decrease in aged discs compared to young discs across all compartments (*Figure 1J*, Wilcoxon p=$5.6 \times 10^{-4}$ ; *Figure 1—figure supplement 4B*).

GAPDH and ACTA2 are two commonly used reference proteins, involved in the metabolic process and the cytoskeletal organisation of cells, respectively. They are expected to be relatively constant between cells and are used to quantify the relative cellular content of tissues (*Barber et al., 2005*). They were detected in all 66 profiles. GAPDH and ACTA2 amounts were significantly correlated with a Pearson correlation coefficient (PCC) of 0.794 (p=$9.3 \times 10^{-9}$) (*Figure 1—figure supplement 4C*), and they were both significantly co-expressed with the detected histone subunits, with

PCCs of 0.785 and 0.636, respectively (*Figure 1K*; *Figure 1—figure supplement 4D*). As expected, expression of the histones, GAPDH, and ACTA2 was not correlated with two core-matrisome proteins, ACAN and COL2A1 (*Figure 1—figure supplement 4E*), whereas ACAN and COL2A1 were significantly co-expressed (*Figure 1—figure supplement 4F*), as expected due to their related regulation of expression and tissue function. Thus, cellularity information can be obtained from proteomic information, and the histone quantification showing reduced cellularity in the aged IVD is consistent with the reported changes (*Rodriguez et al., 2011*; *Sakai et al., 2012*).

## PCA captures the information content distinguishing age and tissue types

To gain a global overview of the data, we performed PCA on a set of 507 proteins selected computationally, allowing maximal capture of valid values, while incurring minimal missing values, followed by imputations (*Figure 2—figure supplement 1*; Materials and methods). The first two principal components (PCs) explained a combined 65.5% of the variance, with 39.9% and 25.6% for the first and second PCs, respectively (*Figure 2A and B*). A support vector machine with polynomial kernel was trained to predict the boundaries: it showed PC1 to be most informative to predict age, with a clear demarcation between the two age groups (*Figure 2A*, vertical boundary), whereas PC2 distinguished disc sample localities, separating the inner compartments (NP, NP/IAF, and IAF) from the OAF (*Figure 2A*, horizontal boundary). PC3 captured only 5.0% of the variance (*Figure 2C & D*), but it distinguished disc level, separating the lowest level (L5/S1) from the rest of the lumbar discs (L3/4 and L4/5) (*Figure 2D*, horizontal boundary). Samples in the upper level (L3/4 and L4/5) appeared to be more divergent, with the aged disc samples deviating from the young ones (*Figure 2D*).

## Top correlated genes with the PCs are insightful of disc homeostasis

To extract the most informative features of the PCA, we performed proteome-wide associations with each of the top three PCs, which accounted for over 70% of total variance, and presented the top 100 most positively and top 100 most negatively correlated proteins for each of the PCs (*Figure 2E–G*). As expected, the correlation coefficients in absolute values were in the order of PC1 > PC2 > PC3 (*Figure 2E–G*). The protein content is presented as non-matrisome proteins (grey colour) and matrisome proteins (coloured) that are subcategorised as previously. For the negatively correlated proteins, the matrisome proteins contributed to PC1 in distinguishing young disc samples, as well as to PC2 for sample location within the disc, but less so for disc level in PC3. Further, the relative composition of the core and non-core matrisome proteins varied between the three PCs, depicting the dynamic ECM requirement and its relevance in aging (PC1), tissue composition within the disc (PC2) and mechanical loading (PC3).

PC1 of young discs identified known chondrocyte markers, CLEC3A (*Lau et al., 2018*) and LECT1/2 (*Zhu et al., 2019*), hedgehog signalling proteins, HHIPL2, HHIP, and SCUBE1 (*Johnson et al., 2012*), and xylosyltransferase-1 (XYLT1), a key enzyme for initiating the attachment of glycosaminoglycan side chains to proteoglycan core proteins (*Silagi et al., 2018*; *Figure 2E*). Most of the proteins that were positively correlated in PC1 were coagulation factors or coagulation related, suggesting enhanced blood infiltration in aged discs. PC2 implicated key changes in molecular signalling proteins (hedgehog, WNT and Nodal) in the differences between the inner and outer disc regions (*Figure 2F*). Notably, PC2 contains heat shock proteins (HSPA1B, HSPA8, HSP90AA1, HSPB1) which are more strongly expressed in the OAF than in inner disc, indicating the OAF is under stress (*Takao and Iwaki, 2002*). Although the correlations in PC3 were much weaker, proteins such as CILP/CILP2, DCN, and LUM were associated with lower disc level.

## ANOVA reveals the principal phenotypes for explaining variability in categories of ECMs

To investigate how age, disc compartment, level, and direction affect the protein profiles, we carried out ANOVA for each of these phenotypic factors for the categories of matrisome and non-matrisome proteins (*Figure 2—figure supplement 1H–J*). In the young discs, the dominant phenotype explaining the variances for all protein categories was disc compartment. It is crucial that each disc compartment (NP, IAF, and OAF) has the appropriate protein composition to function correctly (*Figure 2—figure supplement 1I*). This also fits the understanding that young healthy discs are axially

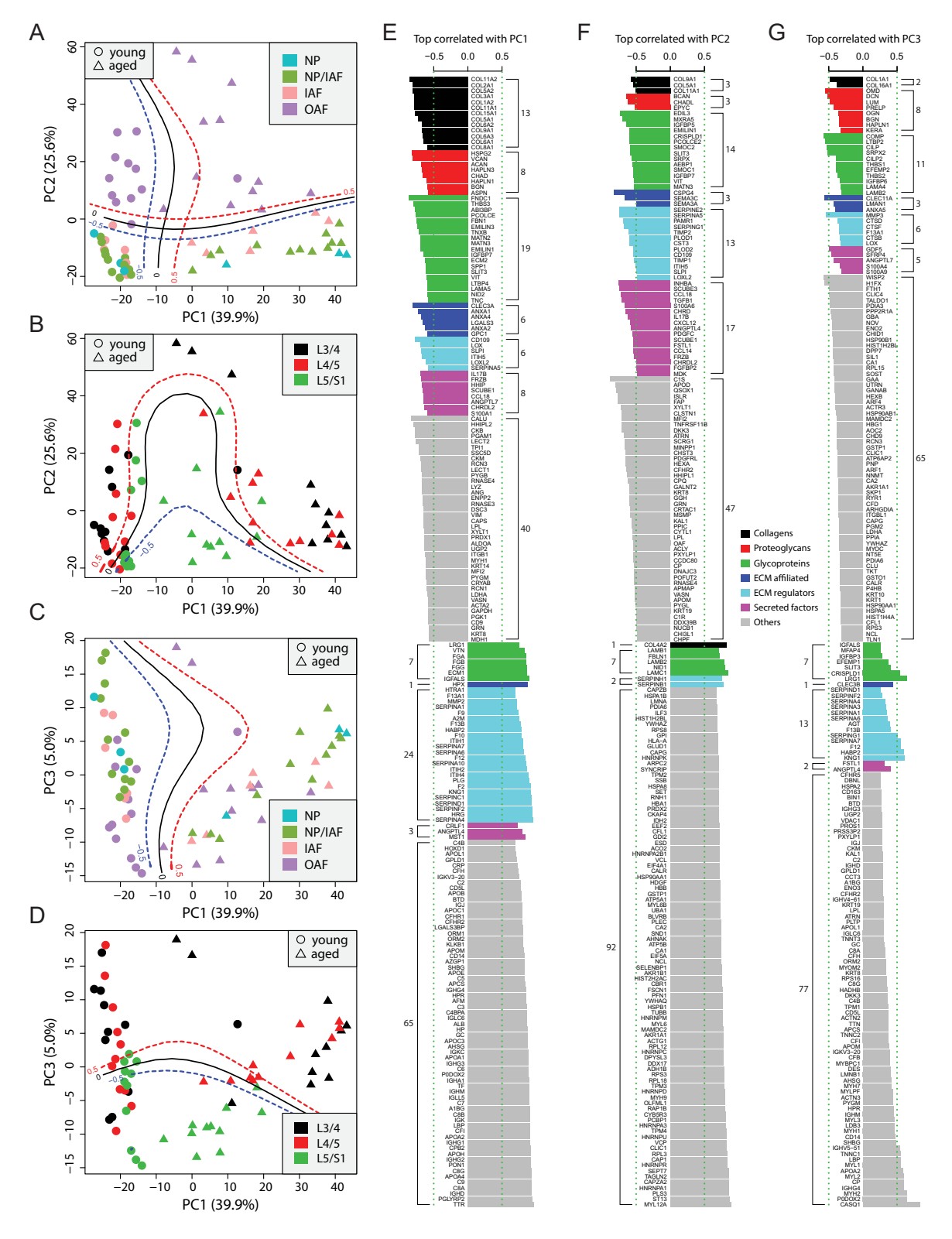

**Figure 2.** Principle component analysis (PCA) of the 66 static spatial profiles based on a set of 507 genes selected by optimal cutoff (see *Figure 2— figure supplement 1A–C*). (**A**) Scatter-plot of PC1 and PC2 colour-coded by compartments, and dot-shaped by age-groups. Solid curves are the support vector machines (SVMs) decision boundaries between inner disc regions (NP, NP/IAF, IAF) and OAF, and dashed curves are soft boundaries for probability equal to ±0.5 and are applied to all plots in this figure. (**B**) Scatter-plot of PC1 and PC2 colour-coded by disc levels. The SVM boundaries are

*Figure 2 continued on next page*

*Figure 2 continued*

trained between L5/S1 and upper levels (L3/4 and L4/5). (C) Scatter-plot of PC1 and PC3, colour-coded by disc compartments. The SVM boundaries are trained between inner disc regions and OAF. (D) Scatter-plot of PC1 and PC3, colour-coded by disc levels. The SVM boundaries are trained between L5/S1 and upper levels (L3/4 and L4/5). (E) Top 100 positively and negatively correlated genes with PC1, colour-coded by ECM categories. (F) Top 100 positively and negatively correlated genes with PC2, colour-coded by extracellular matrix (ECM) categories. (G) Top 100 positively and negatively correlated genes with PC3, colour-coded by ECM categories.

The online version of this article includes the following figure supplement(s) for figure 2:

**Figure supplement 1.** Selection of genes for performing principle component analysis (PCA) and results of ANOVA on assessing global data variability.

---

symmetric and do not vary across disc levels. In aged discs, compartment is still relevant for non-matrisome proteins and collagen, but disc level and directions become influential for other protein categories, which is consistent with variations in mechanical loading occurring in the discs with ageing and degeneration (*Figure 2—figure supplement 1J*). In the combined (young and aged) disc samples, age was the dominant phenotype across major matrisome categories, while compartment best explained the variance in non-matrisome, reflecting the expected changes in cellular (non-matrisome) and structural (matrisome) functions of the discs (*Figure 2—figure supplement 1H*). This guided us to analyse the young and aged profiles separately, before performing cross-age comparisons.

## The high-resolution spatial proteome of young and healthy discs

PCA of the 33 young profiles showed a distinctive separation of the OAF from the inner disc regions on PC1 (upper panel of *Figure 3A*). In PC2, the lower level L5/S1 could generally be distinguished from the upper lumbar levels (lower panel of *Figure 3A*). The detected proteome of the young discs (*Figure 3B*) accounts for 9.2% of the human proteome (or 1883 out of the 20,368 on UniProt). We performed multiple levels of pairwise comparisons (summarised in *Figure 3—figure supplement 1A*) to detect proteins associated with individual phenotypes, using three approaches (see Materials and methods): statistical tests; proteins detected in one group only; or proteins using a fold-change threshold. We detected a set of 671 DEPs (*Supplementary file 2*) (termed the 'variable set'), containing both matrisome and non-matrisome proteins (*Figure 3D*), and visualised in a heatmap (*Figure 3—figure supplement 2*), with identification of four modules (Y1-Y4).

## Protein modules show lateral and anteroposterior trends

To investigate how the modules are associated with disc components, we compared their protein expression profiles along the lateral and anteroposterior axes. The original $\log_2$(LFQ) values were transformed to z-scores to be on the same scale. Proteins of the respective modules were superimposed on the same charts, disc levels combined or separated (*Figure 3E–H*; *Figure 3—figure supplement 3*). Module Y1 is functionally relevant to NP, containing previously reported NP and novel markers KRT19, CD109, KRT8 and CHRD (*Anderson et al., 2002*), CHRDL2, SCUBE1 (*Johnson et al., 2012*), and CLEC3B. Proteins levels in Y1 are lower on one side of the OAF, increases towards the central NP, before declining towards the opposing side, forming a concave pattern in both lateral and anteroposterior directions, with a shoulder drop occurring between IAF and OAF (*Figure 3E*).

Module Y2 was enriched in proteins for ECM organisation (FDR q = $8.8 \times 10^{-24}$); Y3 was enriched for proteins involved in smooth muscle contraction processes (FDR q = $8.96 \times 10^{-19}$); and Y4 was enriched for proteins of the innate immune system (FDR q = $2.93 \times 10^{-20}$). Interestingly, these modules all showed a tendency for convex patterns with upward expression toward the OAF regions, in both lateral and anteroposterior directions, in all levels, combined or separated (*Figure 3F–H*; *Figure 3—figure supplement 3B–D*). The higher proportions of ECM, muscle contraction, and immune system proteins in the OAF are consistent with the contractile function of the AF (*Nakai et al., 2016*), and with the NP being avascular and 'immune-privileged' in a young homeostatic environment (*Sun et al., 2020*).

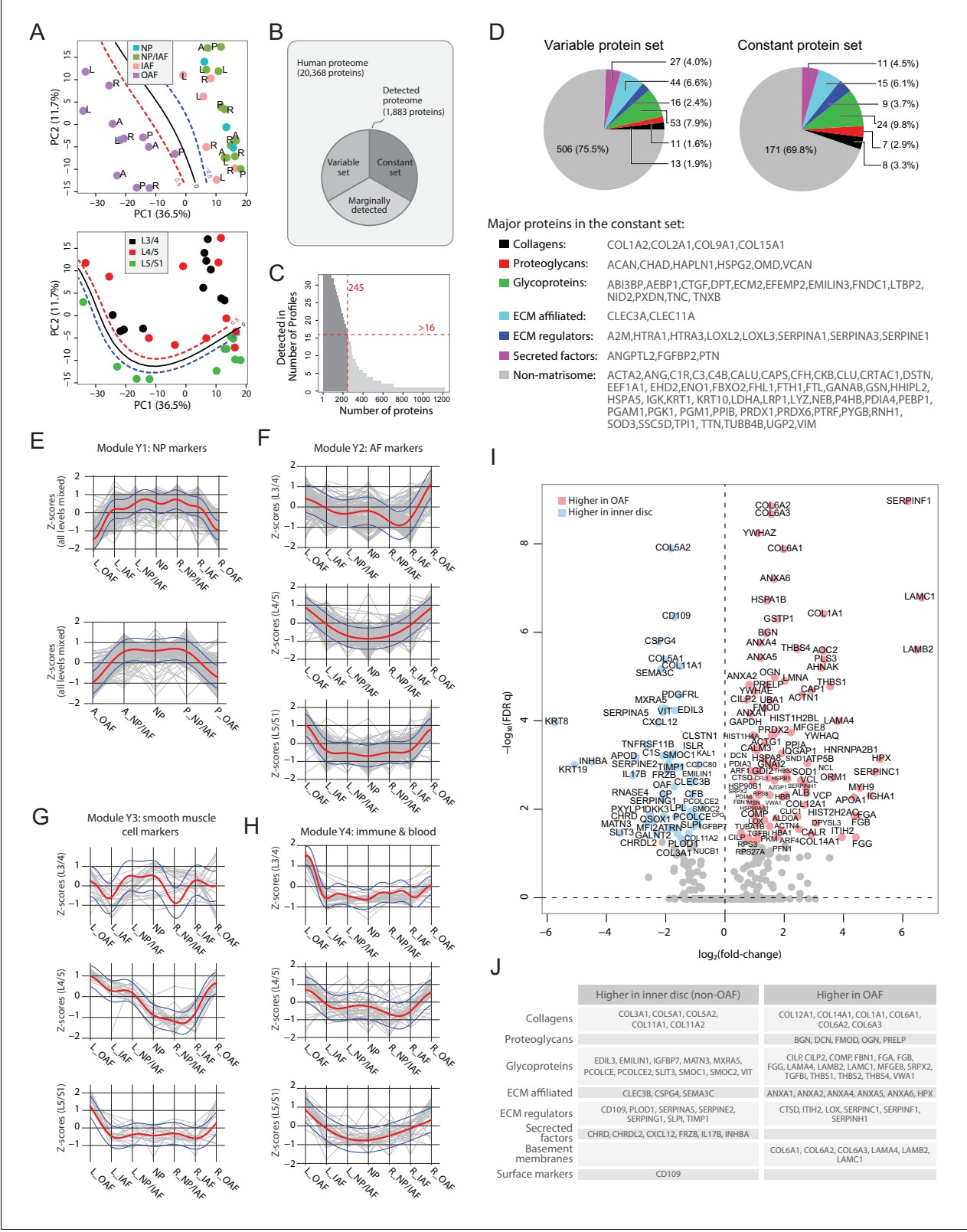

**Figure 3.** Delineating the young non-degenerated cadaveric discs' static spatial proteome. (**A**) Principal component analysis (PCA) plot of all 33 young profiles. Curves in the upper panel show the support vector machine (SVM) boundaries between the OAF and inner disc regions, those in the lower panel separate the L5/S1 disc from the upper disc levels. L, left; R, right; A, anterior; P, posterior. (**B**) A schematic illustrating the partitioning of the detected human disc proteome into variable and constant sets. (**C**) A histogram showing the distribution of non-DEPs in terms of their detected

*Figure 3 continued on next page*

*Figure 3 continued*

frequencies in the young discs. Only 245 non-DEP proteins were detected in over 16 profiles, which is thus defined to be the constant set; while the remaining ~1,000 proteins were considered marginally detected. (D) Piecharts showing the extracellular matrix (ECM) compositions in the variable (left) and constant (right) sets. The constant set proteins that were detected in all 33 young profiles are listed at the bottom. (E) Normalised expression (Z-scores) of proteins in the young module Y1 (NP signature) laterally (top panel) and anteroposteriorly (bottom panel), for all three disc levels combined. The red curve is the Gaussian Process Estimation (GPE) trendline, and the blue curves are one standard deviation above or below the trendline. (F) Lateral trends of module Y2 (AF signature) for each of the three disc levels. (G) Lateral trends of module Y3 (Smooth muscle cell signature) for each of the three disc levels. (H) Lateral trends of module Y4 (Immune and blood) for each of the three disc levels. (I) Volcano plot of differentially expressed proteins (DEPs) between OAF and inner disc (an aggregate of NP, NP/IAF, IAF), with coloured dots representing DEPs. (J) A functional categorisation of the DEPs in (I).

The online version of this article includes the following figure supplement(s) for figure 3:

**Figure supplement 1.** Additional comparisons within the young samples.
**Figure supplement 2.** Heatmap of differentially expressed proteins (DEPs) and protein modules.
**Figure supplement 3.** Modular trends along the lateral or anteroposterior axes in the young non-degenerated discs.

## Inner disc regions are characterised with NP markers

The most distinctive pattern on the PCA is the separation of the OAF from the inner disc, with 99 proteins expressed higher in the OAF and 55 expressed higher in the inner disc (*Figure 3I,J*). Notably, OAF and inner disc contained different types of ECM proteins. The inner disc regions were enriched in collagens (COL3A1, COL5A1, COL5A2, and COL11A2), matrillin (MATN3), and proteins associated with ECM synthesis (PCOLCE) and matrix remodelling (MXRA5). We also identified in the inner disc previously reported NP markers (KRT19, KRT8) (*Risbud et al., 2015*), in addition to inhibitors of WNT (FRZB, DKK3) and BMP (CHRD, CHRDL2) signalling (*Figure 3I,J*). Of note, FRZB and CHRDL2 were recently shown to have potential protective characteristics in osteoarthritis (*Ji et al., 2019*). The TGFβ pathway appears to be suppressed in the inner disc where antagonist CD109 (*Bizet et al., 2011*; *Li et al., 2016*) is highly expressed, and the TGFβ activity indicator TGFBI is expressed higher in the OAF than in the inner disc.

## The OAF signature is enriched with proteins characteristic of tendon and ligament

The OAF is enriched with various collagens (COL1A1, COL6A1/2/3, COL12A1 and COL14A1), basement membrane (BM) proteins (LAMA4, LAMB2 and LAMC1), small leucine-rich proteoglycans (SLRP) (BGN, DCN, FMOD, OGN, PRELP), and BM-anchoring protein (PRELP) (*Figure 3I,J*). Tendon-related markers such as thrombospondins (THBS1/2/4) (*Subramanian and Schilling, 2014*) and cartilage intermediate layer proteins (CILP/CILP2) are also expressed higher in the OAF. Tenomodulin (TNMD) was exclusively expressed in 9 of the 12 young OAF profiles, and not in any other compartments (*Supplementary file 2*). This fits a current understanding of the AF as a tendon/ligament-like structure (*Nakamichi et al., 2018*). In addition, the OAF was enriched in actin-myosin (*Figure 3I*), suggesting a role of contractile function in the OAF, and in heat-shock proteins (HSPA1B, HSPA8, HSPB1, HSP90B1, HSP90AA1), suggesting a stress response to fluctuating mechanical loads.

## Spatial proteome enables clear distinction between IAF and OAF

We sought to identify transitions in proteomic signatures between adjacent compartments. The NP and NP/IAF protein profiles were highly similar (*Figure 3—figure supplement 1B*). Likewise, NP/IAF and IAF showed few DEPs, except COMP which was expressed higher in IAF (*Figure 3—figure supplement 1C*). OAF and NP (*Figure 3—figure supplement 1R*) and OAF and NP/IAF (*Figure 3—figure supplement 1O*) showed overlapping DEPs, consistent with NP and NP/IAF having highly similar protein profiles, despite some differences in the anteroposterior direction (*Figure 3—figure supplement 1P & Q*). The clearest boundary within the IVD, between IAF and OAF, was marked by a set of DEPs, of which COL5A1, SERPINA5, MXRA5 were enriched in the IAF, whereas LAMB2, THBS1, CTSD typified the OAF (*Figure 3—figure supplement 1D*). These findings agreed with the modular patterns (*Figure 3E–H*).

## The constant set represents the baseline proteome among structures within the young disc

Of the 1,880 proteins detected, 1,204 proteins were not found to vary with respect to the phenotypic factors. The majority of these proteins were detected in few profiles (*Figure 3B*) and were not used in the comparisons. We set a cutoff for a detection in >1/2 of the profiles to prioritise a set of 245 proteins, hereby referred to as the 'constant set' (*Figure 3C*). Both the variable and the constant sets contained high proportions of ECM proteins (*Figure 3D*). Amongst the proteins in the constant set that were detected in all 33 young profiles were known protein markers defining a young NP or disc, including COL2A1, ACAN, and A2M (*Risbud et al., 2015*). Other key proteins in the constant set included CHAD, HAPLN1, VCAN, HTRA1, CRTAC1, and CLU. Collectively, these proteins showed the common characteristics shared by compartments of young discs, and they, alongside the variable set, form the architectural landscape of the young disc.

## Diverse changes in the spatial proteome with ageing

PCA was used to identify compartmental, directional, and level patterns for the aged discs (*Figure 4A*). Albeit less clear than for the young discs (*Figure 3A*), the OAF could be distinguished from the inner disc regions on PC1, explaining 46.7% of the total variance (*Figure 4A*). PC2 showed a more distinct separation of signatures from lumbar disc levels L5/S1 to the upper disc levels (L4/L5 and L3/4), accounting for 21.8% of the total variance (*Figure 4A*).

## Loss of the NP signature from inner disc regions

As with the young discs, we performed a series of comparative analyses (*Figure 4C*; *Figure 4—figure supplement 1A–H*; *Supplementary file 3*). Detection of DEPs between the OAF and the inner disc (*Figure 4B*), showed that 100 proteins were expressed higher in the OAF, similar to young discs (*Figure 3C*). However, in the inner regions, only nine proteins were significantly expressed higher, in marked contrast to the situation in young discs. Fifty-five of the 100 DEPs in the OAF region overlapped in the same region in the young discs, but only 3 of the 9 DEPs in the inner region were identified in the young disc; indicating changes in both regions but more dramatic in the inner region. This suggests that ageing and associated changes may have initiated at the centre of the disc. The typical NP markers (KRT8/19, CD109, CHRD, CHRDL2) were not detectable as DEPs in the aged disc; but CHI3L2, A2M and SERPING1 (*Figure 4B*), which have known roles in tissue fibrosis and wound healing (*Lee et al., 2011*; *Naveau et al., 1994*; *Wang et al., 2019a*), were detected uniquely in the aged discs.

## Shift of ECM composition and cellular responses in the outer AF

A comparative analysis of the protein profiles indicated that the aged OAF retained 55% of the proteins of a young OAF. These changes are primarily reflected in the class of SLRPs (BGN, DCN, FMOD, OGN, and PRELP), and glycoproteins such as CILP, CILP2, COMP, FGA/B, and FGG. From a cellular perspective, 45 proteins enriched in the aged OAF could be classified under 'responses to stress' (FDR q = $1.86 \times 10^{-7}$; contributed by CAT, PRDX6, HSP90AB1, EEF1A1, TUBB4B, P4HB, PRDX1, HSPA5, CRYAB, HIST1H1C), suggesting OAF ECM and cells are responding to a changing environment such as mechanical loading and other stress factors.

## Convergence of the inner disc and outer regions in aged discs

To map the relative changes between inner and outer regions of the aged discs, we performed a systematic comparison between compartments (*Figure 4—figure supplement 1A–D*). The most significant observation was a weakening of the distinction between IAF and OAF that was seen in young discs, with only 17 DEPs expressed higher in the OAF of the aged disc (*Figure 4—figure supplement 1C*). More differences were seen when we included the NP in the comparison with OAF (*Figure 4—figure supplement 1D*), indicating some differences remain between inner and outer regions of the aged discs. While the protein profiles of the NP and IAF were similar, with no detectable DEPs (*Figure 4—figure supplement 1A & B*), their compositions shared more resemblance with the OAF. These progressive changes of the protein profiles and DEPs between inner and outer compartments suggest the protein composition of the inner disc compartments becomes more

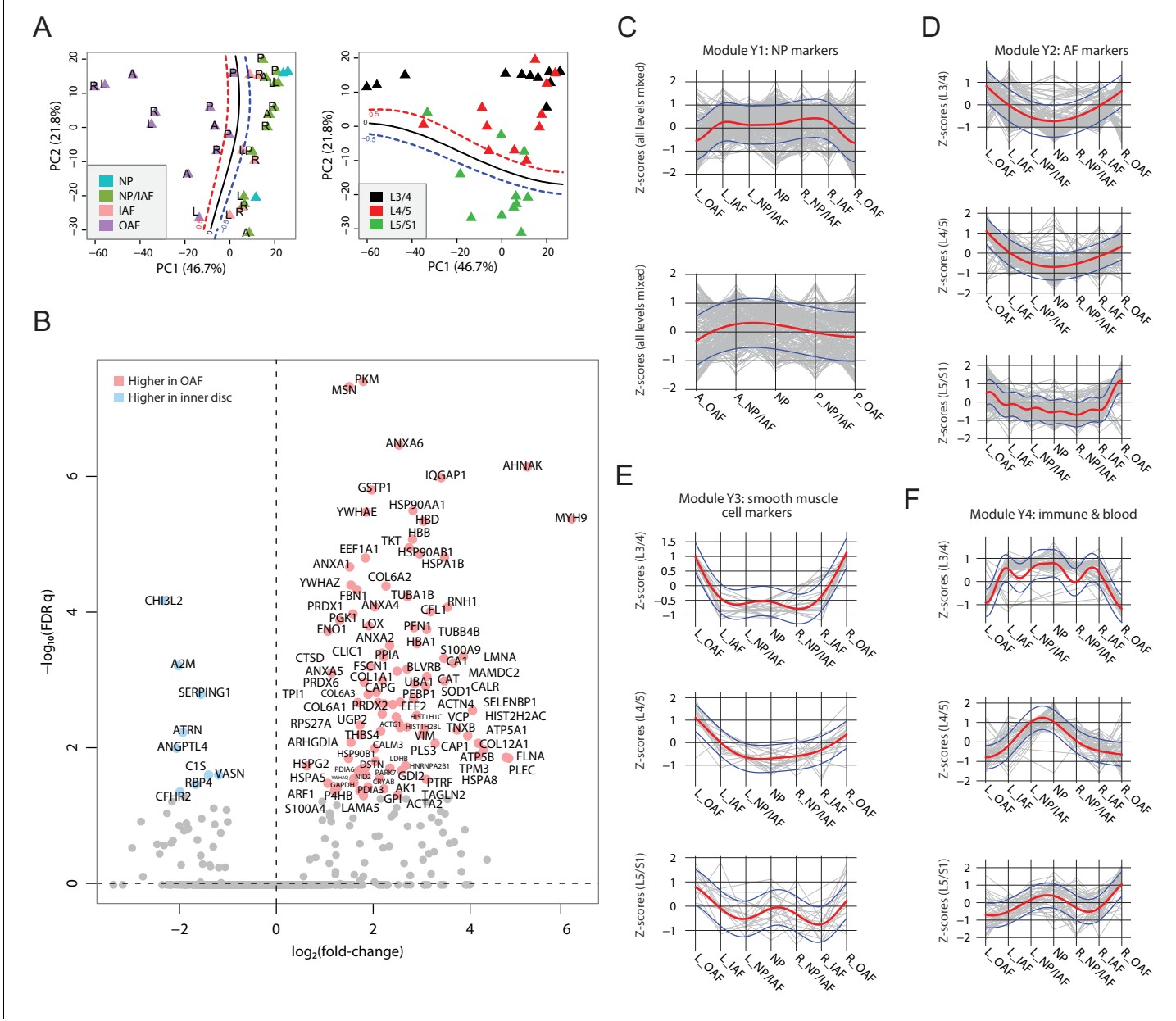

**Figure 4.** Characterisation of the aged cadaveric discs' static spatial proteome. (A) Principal component analysis (PCA) plot of all the aged profiles on PC1 and PC2, colour-coded by compartments. Curves in the left panel show the support vector machine (SVM) boundaries between OAF and inner disc; those in the right panel separate the L5/S1 disc from the upper disc levels. Letters on dots indicate directions: L, left; R, right; A, anterior; P, posterior. (B) Volcano plot showing the differentially expressed proteins (DEPs) between the OAF and inner disc (an aggregate of NP, NP/IAF, and IAF), with the coloured dots representing statistically significant (FDR < 0.05) DEPs. (C) Using the same four modules identified in young samples, we determined the trend for these in the aged samples. Locational trends of module Y1 showing higher expression in the inner disc, albeit they are more flattened than in the young disc samples. Top panel shows left to right direction and bottom panel shows anterior to posterior direction. The red curve is the Gaussian Process Estimation (GPE) trendline, and the blue curves are one standard deviation above or below the trendline. This also applies to (D), (E), and (F). (D) Lateral trends for module Y2 in the aged discs. (E) Lateral trends for module Y3 in the aged discs. (F) Lateral trends for module Y4 in the aged discs.

The online version of this article includes the following figure supplement(s) for figure 4:

**Figure supplement 1.** Differentially expressed proteins (DEPs) between different compartments or levels and spatial trends of protein modules in the aged discs.

similar to the OAF with ageing, with the greatest changes in the inner regions. This further supports a change initiating from the inner region of the discs with ageing.

## Changes in young module patterns reflect convergence of aged disc compartments

We investigated protein composition in the young disc modules (Y1-Y4) across the lateral and anteroposterior axes in the aged discs (*Figure 4C–F*). For module Y1 that consists of proteins defining the NP region, the distinctive concave pattern has flattened along both axes, but more so for the anteroposterior direction where the clear interface between IAF and OAF was lost (*Figure 4C*; *Figure 4—figure supplement 1I*). Similarly for modules Y2 and Y3, which consist of proteins defining the AF region, the trends between inner and outer regions of the disc have changed such that the patterns become more convex, with a change that is a continuum from inner to outer regions (*Figure 4D,E*; *Figure 4—figure supplement 1J,K*). These changes in modules Y1-3 in the aged disc further illustrate the convergence of the inner and outer regions, with the NP/IAF becoming more OAF-like. For Y4, the patterns along the lateral and anteroposterior axes were completely disrupted (*Figure 4F*; *Figure 4—figure supplement 1L*). As Y4 contains proteins involved in vascularity and inflammatory processes (*Supplementary file 5*), these changes indicate disruption of cellular homeostasis in the NP.

## Disc level variations reflect spatial and temporal progression of disc changes

The protein profiles of the aged disc levels, consistent with the PCA findings, showed similarity between L3/4 and L4/5 (*Figure 4—figure supplement 1E*), but, in contrast to young discs, differences between L5/S1 and L4/5 (*Figure 4—figure supplement 1F*), and more marked differences between L5/S1 and L3/4 (*Figure 4—figure supplement 1G,H*). Overall, the findings from PCA, protein profiles (*Figure 2B–D*), and MRI (*Figure 1C*) agree. As compared to young discs, the more divergent differences across the aged disc levels potentially reflect progressive transmission from the initiating disc to the adjacent discs with ageing. To further investigate the aetiologies underlying IDD, cross-age comparisons are needed.

## Aetiological insights uncovered by young/aged comparisons

Next, we performed extensive pair-wise comparisons between the young and aged samples under a defined scheme (*Figure 5—figure supplement 1A*; *Supplementary file 4*). First, we compared all 33 young samples with all 33 aged samples, which identified 169 DEPs with 104 expressed higher in the young and 65 expressed higher in the aged discs (*Figure 5A*). A simple gene ontology (GO) term analysis showed that the most important biological property for a young disc is structural integrity, which is lost in aged discs (*Figure 5B*; *Supplementary file 5*). The protein classes most enriched in the young discs were related to cartilage synthesis, chondrocyte development, and ECM organisation (*Figure 5B*). The major changes in the aged discs relative to young ones, were proteins involved in cellular responses to an ageing environment, including inflammatory and cellular stress signals, progressive remodelling of disc compartments, and diminishing metabolic activities (*Figure 5C*; *Supplementary file 5*).

## Inner disc regions present with most changes in ageing

For young versus old discs, we compared DEPs of the whole disc (*Figure 5A*) with those from the inner regions (*Figure 5D*) and OAF (*Figure 5E*) only. Seventy-five percent (78/104) of the downregulated DEPs (*Figure 5F*) were attributed to the inner regions, and only 17% (18/104) were attributed to the OAF (*Figure 5F*). Similarly, 65% (42/65) of the upregulated DEPs were solely contributed by inner disc regions, and only 18.4% (12/65) by the OAF. Only 5 DEPs were higher in the OAF, while 49 were uniquely upregulated in the inner discs (*Figure 5G*). The key biological processes in each of the compartments are highlighted in *Figure 5F and G*.

## The perturbed biological processes in the aged discs

Expression of known NP markers was reduced in aged discs, especially proteins involved in the ECM and its remodelling, where many of the core matrisome proteins essential for the structural function

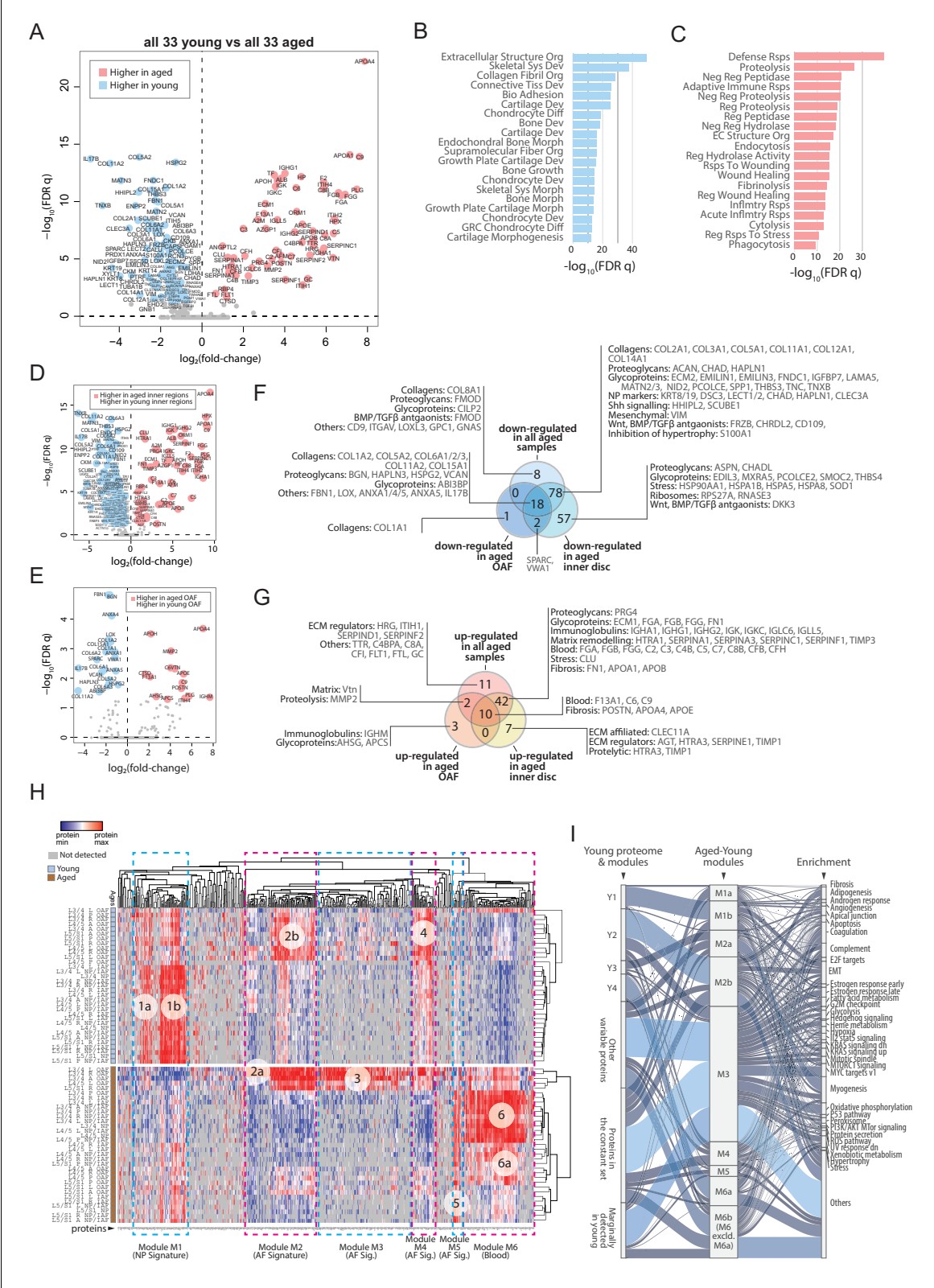

**Figure 5.** Comparisons between young and aged static spatial proteomes. (**A**) Volcano plot showing the differentially expressed proteins (DEPs) between all the 33 young and 33 aged profiles. Coloured dots represent statistically significant DEPs. (**B**) Gene ontology (GO) term enrichment of DEPs higher in young profiles. (**C**) GO term enrichment of DEPs higher in aged profiles. Full names of GO terms in (**B** and **C**) are listed in ***Supplementary file 5***. (**D**) Volcano plot showing DEPs between aged and young inner disc regions. (**E**) Volcano plot showing DEPs between aged and young OAF. (**F**) Venn

*Figure 5 continued on next page*

*Figure 5 continued*

diagram showing the partitioning of the young/aged DEPs that were downregulated in aged discs, into contributions from inner disc regions and OAF. (**G**) Venn diagram showing the partitioning of the young/aged DEPs that were upregulated in aged discs, into contributions from inner disc regions and OAF. (**H**) A heat map showing proteins expressed in all young and aged disc, with the identification of 6 modules (module 1: higher expression in young inner disc regions, modules 2 and 4: higher expression in young OAF, module 3: highly expressing in aged OAF, module 5: higher expression across all aged samples, and module 6: higher expression in aged inner disc, and some OAF). (**I**) An alluvial chart showing the six modules identified in (**H**) and their connections to the previously identified four modules and constant set in the young reference proteome; as well as their connections to enriched GO terms.

The online version of this article includes the following figure supplement(s) for figure 5:

**Figure supplement 1.** Comparisons of young and aged proteomes.

of the NP were less abundant or absent. While this is consistent with previous observations (*Feng et al., 2006*), of interest was the presence of a set of protein changes that were also seen in the OAF, which was also rich in ECM and matrix remodelling proteins (HTRA1, SERPINA1, SERPINA3, SERPINC1, SERPINF1, and TIMP3) and proteins involved in fibrotic events (FN1, POSTN, APOA1, APOB), suggesting these changes are occurring in the ageing IVDs (*Figure 5F,G*).

Proteins associated with cellular stress are decreased in the aged inner disc, with functions ranging from molecular chaperones needed for protein folding (HSPB1, HSPA1B and HSPA9) to modulation of oxidative stress (SOD1) (*Figure 5F*). SOD1 has been shown to become less abundant in the aged IVD (*Hou et al., 2014*) and in osteoarthritis (*Scott et al., 2010*). HSPB1 is cytoprotective and a deficiency is associated with inflammation reported in degenerative discs (*Wuertz et al., 2012*). We found an increased concentration of clusterin (CLU) (*Figure 5G*), an extracellular chaperone that aids the solubilisation of misfolded protein complexes by binding directly and preventing protein aggregation (*Trougakos, 2013*; *Wyatt et al., 2009*), and also has a role in suppressing fibrosis (*Peix et al., 2018*).

Inhibitors of WNT (DKK3 and FRZB), and antagonists of BMP/TGFβ (CD109, CHRDL2, DCN FMOD, INHBA and THBS1) signalling were decreased or absent in the aged inner region (*Figure 5F, G*), consistent with the reported upregulation of these pathways in IDD (*Hiyama et al., 2010*) and its closely related condition osteoarthritis (*Leijten et al., 2013*). Targets of hedgehog signalling (HHIPL2 and SCUBE1) were also reduced (*Figure 5F*), consistent with SHH's key roles in IVD development and maintenance (*Rajesh and Dahia, 2018*). TGFβ signalling is a well-known pathway associated with fibrotic outcomes. WNT is known to induce chondrocyte hypertrophy (*Dong et al., 2006*), that can be enhanced by a reduction in S100A1 (*Figure 5F*), a known inhibitor of chondrocyte hypertrophy (*Saito et al., 2007*).

To gain an overview of the disc compartment variations between young and aged discs, we followed the same strategy as in *Figure 3—figure supplement 2* to aggregate three categories of DEPs in all 23 comparisons (*Figure 5—figure supplement 1A*) and created a heatmap from the resulting 719 DEPs. This allowed us to identify six major protein modules (*Figure 5H*). A striking feature is module 6 (M6), which is enriched for proteins involved in the complement pathway (GSEA Hallmark FDR q = $4.9 \times 10^{-14}$) and angiogenesis (q = $2.3 \times 10^{-3}$). This module contains proteins that are all highly expressed in the inner regions of the aged disc, suggesting the presence of blood. M6 also contains the macrophage marker CD14, which supports this notion.

We visualised the relationship between the young (Y1-4) and young/aged (M1-6) modules using an alluvial chart (*Figure 5I*). Y1 corresponds primarily to M1b that is enriched with fibrosis, angiogenesis, apoptosis, and EMT (epithelial to mesenchymal transition) proteins. Y2 seems to have been deconstructed into three M modules (M2b > M3 > M4). M2b and M3 contain proteins linked to heterogeneous functions, while proteins in M4 are associated with myogenesis and cellular metabolism, but also linked to fibrosis and angiogenesis. Y3 primarily links to M2a with a strong link to myogenesis, and mildly connects with M3 and M4. Y4 has the strongest connection with M6a, which is linked to coagulation. Both the variable and the constant sets of the young disc were also changed in ageing. In the constant set, there is a higher tendency for a decrease in ECM-related proteins, and an increase in blood and immune related proteins with ageing, that may reflect an erosion of the foundational proteome, and infiltration of immune cells (*Figure 5—figure supplement 1L*).

In all, the IVD proteome showed that with ageing, activities of the SHH pathways were decreased, while those of the WNT and BMP/TGFβ pathways, EMT, angiogenesis, fibrosis, cellular stresses, and chondrocyte hypertrophy-like events were increased.

## Comparison between transcriptomic and proteomic data

The proteome reflects both current and past transcriptional activities. To investigate upstream cellular and regulatory activities, we obtained transcriptome profiles from two IVD compartments (NP and AF) and two sample states (young, scoliotic but non-degenerated, YND; aged individuals with clinically diagnosed IDD, AGD) (*Table 1*). The transcriptome profiles of YND and AGD are similar to the young and aged disc proteome samples, respectively. After normalisation (*Figure 6—figure supplement 1A*) and hierarchical clustering, we found patterns reflecting relationships among IVD compartments and ages/states (*Figure 6—figure supplement 1B–D*). PCA of the transcriptome profiles showed that PC1 captured age/state variations (*Figure 6A*) and PC2 captured the compartment (AF or NP) differences, with a high degree of similarity to the proteomic PCA that explained 65.0% of all data variance (*Figure 2A*).

## Transcriptome shows AF-like characteristics of the aged/degenerated NP

We compared the transcriptome profiles of different compartments and age/state-groups (*Figure 6—figure supplement 1E–H*). We detected 88 DEGs (Materials and methods) between young AF and young NP samples (*Figure 6—figure supplement 1E*),in which 39 were more abundant in young NP (including known NP markers *CD24*, *KRT19*) and 49 were more abundant in young AF, including the AF markers, *COL1A1*, and *THBS1*. In the AGD samples, 11 genes differed between AF and NP (*Figure 6—figure supplement 1F*), comparable to the proteome profiles (*Figure 4C*). Between the YND and AGD AF, there were 45 DEGs, with *COL1A1* and *MMP1* more abundant in YND and *COL10A1*, WNT signalling (*WIF1*, *WNT16*), inflammatory (*TNFAIP6*, *CXCL14*, *IL11*), and fibrosis-associated (*FN1*, *CXCL14*) genes more abundant in AGD (*Figure 6—figure supplement 1G*). The greatest difference was between YND and AGD NP, with 216 DEGs (*Figure 6—figure supplement 1H*), with a marked loss of NP markers (*KRT19*, *CD24*), and gain of AF (*THBS1*, *DCN*), proteolytic (*ADAMTS5*), and EMT (*COL1A1*, *COL3A1*, *PDPN*, *NT5E*, *LTBP1*) markers with age. Again, consistent with the proteomic findings, the most marked changes are in the NP, with the transcriptome profiles becoming AF-like.

## Concordance between transcriptome and proteome profiles

We partitioned the DEGs between the YND and AGD into DEGs for individual compartments (*Figure 6B*). The transcriptomic (*Figure 6B*) Venn diagram was very similar to the proteomic one (*Figure 5F–G*). For example, WNT/TGFβ antagonists and ECM genes were all downregulated with ageing/degeneration, while genes associated with stress and ECM remodelling were more common. When we directly compared the transcriptomic DEGs and proteomic DEPs across age/states and compartments (*Figure 6C–F*), we observed strong concordance between the two types of datasets for a series of markers. In the young discs, concordant markers included *KRT19* and *KRT8*, *CHRDL2*, *FRZB*, and *DKK3* in the NP, and *COL1A1*, *SERPINF1*, *COL14A1*, and *THBS4* in the AF (*Figure 6C*). In the AGD discs, concordant markers included *CHI3L2*, *A2M* and *ANGPTL4* in the NP and *MYH9*, *HSP90AB1*, *HBA1*, and *ACTA2* in the AF (*Figure 6D*). A high degree of concordance was also observed when we compared across age/states for the AF (*Figure 6E*) and NP (*Figure 6F*).

Despite the transcriptomic samples having diagnoses (scoliosis for YND and IDD for AGD), whereas the proteome samples were cadaver samples with no reported diagnosis of IDD, the changes detected in the transcriptome profiles substantially support the proteomic findings. A surprising indication from the transcriptome was the increased levels of *COL10A1* (*Lu et al., 2014*), *BMP2* (*Grimsrud et al., 2001*), *IBSP*, defensin beta-1 (*DEFB1*), *ADAMTS5*, pro-inflammatory (*TNFAIP6*, *CXCL*) and proliferation (*CCND1*, *IGFBP*) genes in the AGD NP (*Figure 6—figure supplement 1E–H*), reaffirming the involvement of hypertrophic-like events (*Melas et al., 2014*) in the aged and degenerated NP.

The genome-wide transcriptomic data included over 20 times more genes per profile than the proteomic data, providing additional biological information about the disc, particularly low

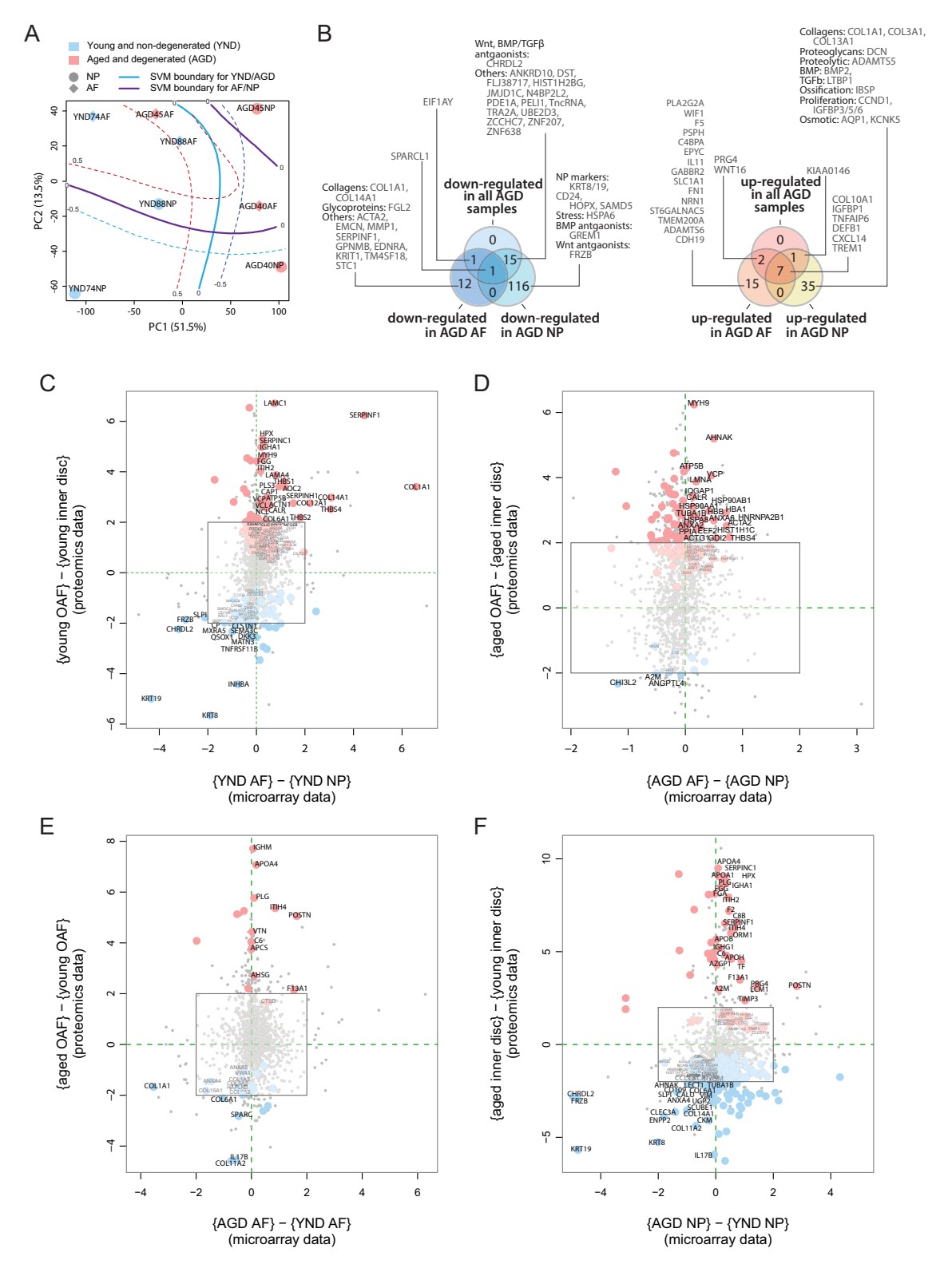

**Figure 6.** Concordance between static spatial proteomic and transcriptome data. (**A**) A principal component analysis (PCA) plot of the eight transcriptomic profiles. Curves represent support vector machine (SVM) boundaries between patient-groups or compartments. (**B**) Venn diagrams showing the partitioning of the young/aged DEGs into contributions from inner disc regions and OAF. Left: downregulated in AGD samples; right: upregulated. (**C**) Transcriptome data from the NP and AF of two young individuals were compared to the proteomic data, with coloured dots

*Figure 6 continued on next page*

*Figure 6 continued*

representing identified proteins also expressed at the transcriptome level. (D) Transcriptome and proteome comparison of aged OAF and NP. (E) Transcriptome and proteome comparison of young and aged OAF. (F) Transcriptome and proteome comparison of young and aged NP.

The online version of this article includes the following figure supplement(s) for figure 6:

**Figure supplement 1.** Microarray transcriptomic data of the NP and AF from four individuals, two of which are young and non-degenerated and the other two aged and degenerated.

abundance proteins, such as transcription factors and surface markers. For example, additional WNT antagonists were *WIF1* (Wnt inhibitory factor) and *GREM1* (*Figure 6B Leijten et al., 2013*). Comparing the YND NP against YND AF or AGD NP (*Figure 6—figure supplement 1E,H*), we identified higher expression of three transcription factors, *T* (brachyury), *HOPX* (homeodomain-only protein homeobox), and *ZNF385B* in the YND NP. Brachyury is a well-known marker for the NP (*Risbud et al., 2015*), and *HOPX* is differentially expressed in mouse NP as compared to AF (*Veras et al., 2020*), and expressed in mouse notochordal NP cells (*Lam, 2013*). Overall, transcriptomic data confirmed the proteomic findings and revealed additional markers.

## The active proteomes in the ageing IVD

The proteomic data up to this point is a static form of measurement (static proteome) and represents the accumulation and turnover of all proteins up to the time of harvest. The transcriptome indicates genes that are actively transcribed, but does not necessarily correlate to translation or protein turnover. Thus, we studied changes in the IVD proteome (dynamic proteome) that would reflect newly synthesised proteins and proteins cleaved by proteases (degradome), and how they relate to the static proteomic and transcriptomic findings reported above.

## Aged or degenerated discs synthesise fewer proteins

We performed *ex vivo* labelling of newly synthesised proteins using the SILAC protocol (*Ong et al., 2002*; *Figure 7A*; Materials and methods) on AF and NP samples from four YND individuals and one AGD individual (*Table 1*). In the SILAC profiles, light isotope-containing signals correspond to the pre-existing unlabelled proteome, and heavy isotope-containing signals to newly synthesised proteins (*Figure 7B*). The ECM compositions in the light isotope-containing profiles (*Figure 7B*, middle panel) are similar to the static proteome samples of the corresponding age groups described above (*Figure 1F*). Although for NP_YND152, the numbers of identified proteins in the heavy profiles are considerably less than NP_YND151 due to a technical issue during sample preparation, it is overall still more similar to NP_YND152 than to other samples (*Figure 7—figure supplement 1B*), indicating that its biological information is still representative of a young NP and the respective AF samples are similar. In contrast, the heavy isotope-containing profiles contained fewer proteins in the AGD than in the YND samples (*Figure 7B*, left panel) and showed variable heavy to light ratio profiles (*Figure 7B*, right panel).

To facilitate comparisons, we averaged the abundance of the proteins detected in the NP or AF for which we had more than one sample, then ranked the abundance of the heavy isotope-containing (*Figure 7C*), light isotope-containing (*Figure 7D*) proteins. The number of proteins newly synthesised in the AGD samples was about half that in the YND samples (*Figure 7C*). This is unlikely to be a technical artefact as the total number of light isotope-containing proteins detected in the AGD samples is comparable to the YND, in both AF and NP (*Figure 7D*), and the difference is again well illustrated in the heavy to light ratios (*Figure 7—figure supplement 1A*).

Reduced synthesis of non-matrisome proteins was found for the AGD samples (GAPDH) as a reference point (dotted red lines) (*Figure 7C and D*; *Figure 7C and E*, grey portions). Of the 68 high abundant non-matrisome proteins in the YND NP compartment that were not present in the AGD NP, 28 are ribosomal proteins (*Figure 7—figure supplement 1C*), suggesting reduced translational activities. This agrees with our earlier findings of cellularity, as represented by histones, in the static proteome (*Figure 1J,K*).

Changes in protein synthesis in response to the cell microenvironment affects the architecture of the disc proteome. To understand how the cells may contribute and respond to the accumulated matrisome in the young and aged disc, we compared the newly synthesised matrisome proteins of

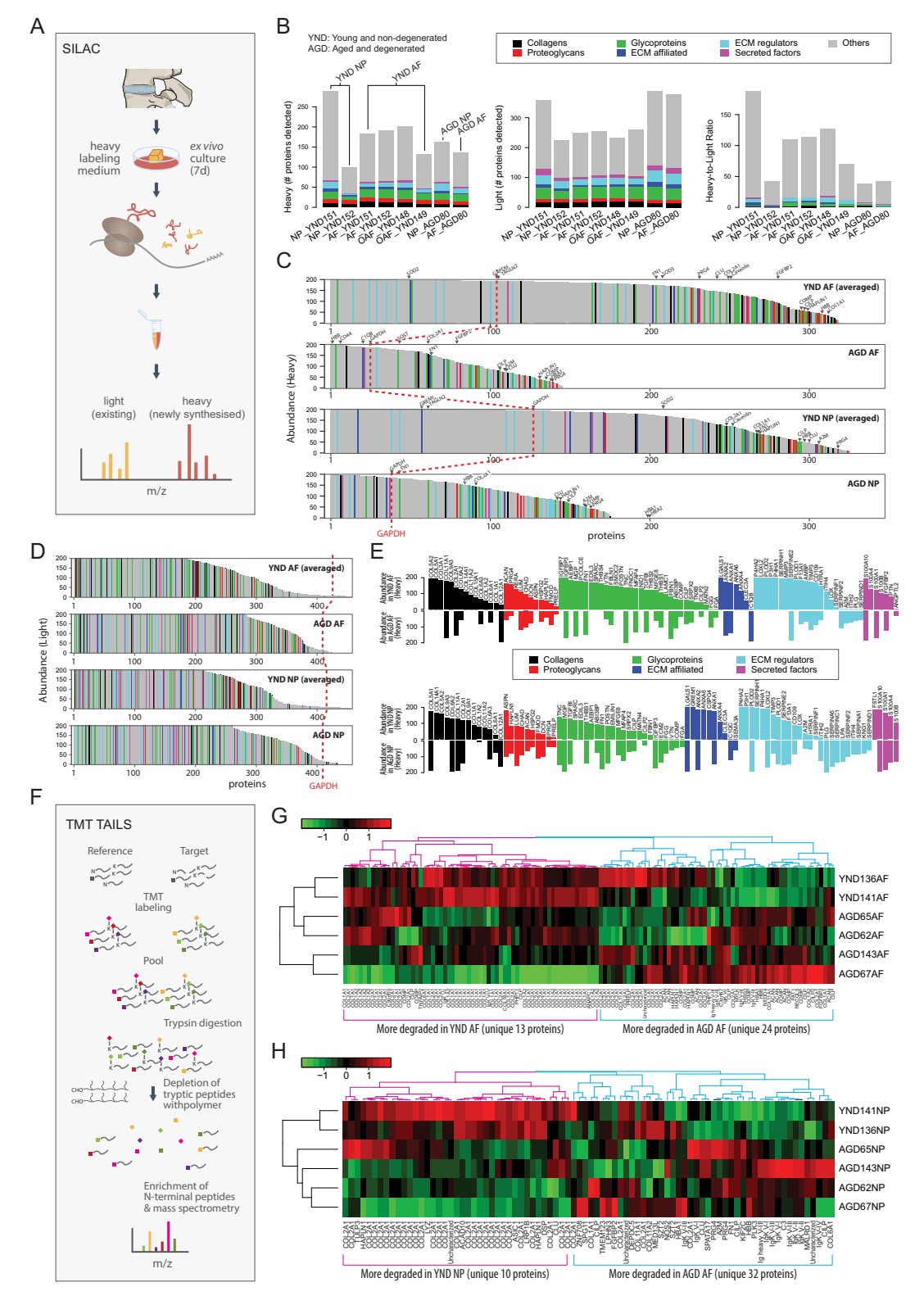

**Figure 7.** The dynamic proteome of the intervertebral disc shows less biosynthesis of proteins and more degradative events in aged tissues. (**A**) Schematic showing pulse-SILAC labelling of *ex-vivo* cultured disc tissues where heavy Arg and Lys are incorporated into newly made proteins (heavy), and pre-existing proteins remaining unlabelled (light). NP and AF tissues from young (n = 3) and aged (n = 1) were cultured for 7 days in hypoxia prior to MS. (**B**) Barcharts showing the number of identified non-matrisome (grey) and matrisome (coloured) existing proteins (middle panel); newly

*Figure 7 continued on next page*

Figure 7 continued

synthesised proteins (left panel), and the heavy/light ratio (right panel) for each of the samples. (C) The quantities of each of the heavy labelled (newly synthesised) proteins identified for each of the four groups were averaged, and then plotted in descending order of abundance. It shows that YND AF and NP synthesise higher numbers of proteins than the AGD AF and NP. The red dotted reference line shows the expression of GAPDH. (D) The quantities of each of the light (existing) proteins identified for each group was averaged, and then plotted in descending order of abundance which shows that there are similar levels of existing proteins in the four pooled samples. (E) The matrisome proteins of (C) were singled out for display. The abundance of these proteins in YND samples were generally higher across all types of matrisome proteins than the AGD, with the exceptions of aged related proteins. (F) Schematic showing the workflow of degradome analysis by N-terminal amine isotopic labelling (TAILS) for the identification of cleaved neo N-terminal peptides. (G) Heatmap showing the identification of cleaved proteins ranked according to tandem mass tag (TMT) isobaric labelling of N-terminal peptides in AF. Data is expressed as the log$_2$(ratio) of N-terminal peptides. (H) Heatmap showing the identification of cleaved proteins ranked according to tandem mass tag (TMT) isobaric labelling of N-terminal peptides in NP. Data is expressed as the log$_2$(ratio) of N-terminal peptides. AGD143 in (G and H) is aged but not degenerated (trauma).

The online version of this article includes the following figure supplement(s) for figure 7:

**Figure supplement 1.** SILAC data.
**Figure supplement 2.** Degradome data.

AGD and YND samples rearranged in order of abundance (*Figure 7E*). More matrisome proteins were synthesised in YND samples across all classes. In YND AF, collagens were synthesised in higher proportions than in AGD AF with the exception of fibril-associated COL12A1 (*Figure 7E*, top panel). Similarly, higher proportions in YND AF were observed for proteoglycans (except FMOD), glycoproteins (except TNC, FBN1, FGG and FGA), ECM affiliated proteins (except C1QB), ECM regulators (except SERPINF2, SERPIND1, A2M, ITIH2, PLG), and secreted factors (except ANGPTL2). Notably, regulators that were exclusively synthesised in young AF are involved in collagen synthesis (P4HA1/2, LOXL2, LOX, PLOD1/2) and matrix turnover (MMP3), with enrichment of protease HTRA1 and protease inhibitors TIMP3 and ITIH in YND AF compared to AGD AF.

In AGD NP, overall collagen synthesis was less than in YND NP (*Figure 7E*, lower panel); however, there was more synthesis of COL6A1/2/3 and COL12A1. Furthermore, AGD NP synthesised more LUM, FMOD, DCN, PRG4, and PRELP proteoglycans than YND NP. Notably, there was less synthesis of ECM-affiliated proteins (except C1QC and SEMA3A) and regulators – particularly those involved in collagen synthesis (P4HA1/2, LOXL2, LOX) – but an increase in protease inhibitors. A number of newly synthesised proteins in AGD NP were similarly represented in the transcriptome data, including POSTN, ITIH2, SERPINC1, IGFBP3, and PLG. Some genes were simultaneously underrepresented in the AGD NP transcriptome and newly synthesised proteins, including hypertrophy inhibitor GREM1 (*Leijten et al., 2013*).

## Proteome of aged or degenerated discs is at a higher degradative state

The degradome reflects protein turnover by identifying cleaved proteins in a sample (*López-Otín and Overall, 2002*). When combined with relative quantification of proteins through the use of isotopic and isobaric tagging and enrichment for cleaved neo amine (N)-termini of proteins before labelled samples are quantified by mass-spectrometry, degradomics is a powerful approach to identify the actual status of protein cleavage *in vivo*.

We employed the well-validated and sensitive terminal amine isotopic labelling of substrates (TAILS) method (*Kleifeld et al., 2010*; *Rauniyar and Yates, 2014*) to analyse and compare six discs from six individuals (two young and non-degenerated, YND; and four aged and/or degenerated, AGD) (*Table 1*; *Figure 7F*; *Kleifeld et al., 2010*; *Rauniyar and Yates, 2014*) using the 6-plex tandem mass tag (TMT)-TAILS (labelling six independent samples and analysed together on the mass spectrometer) (*Figure 7F*). Although shotgun proteomics is intended to identify the proteome components, N-terminome data is designed to identify the exact cleavage site in proteins that also evidence stable cleavage products *in vivo*.

Here, TAILS identified 123 and 84 cleaved proteins in the AF and NP disc samples, respectively. Performing hierarchical clustering on the data we found that the two YND samples (136 and 141; *Table 1*) tend to cluster together in both AF and NP (*Figure 7G,H*; *Figure 7—figure supplement 2A,B*). Interestingly, the trauma sample AGD143 (53 year male), who has no known IDD diagnosis, tend to cluster with other clinically diagnosed AGD samples, in both AF and NP. This might be

because AGD143 has unreported degeneration or ageing is a dominant factor in degradome signals.

We identified two protein/peptide modules in the AF (*Figure 7G*), corresponding to more degradation/cleaving in YND AF (magenta) and AGD AF (blue), respectively. There are only 13 unique proteins for proteins/peptides more degraded in the YND AF, the most common of which is COL1A1/2, followed by COL2A1. In comparison, the module corresponding to more degradation in AGD AF recorded 24 unique proteins, 7 (CILP, CILP2, COL1A1, COMP, HBA1, HBB, PRELP) of which are in strong overlap ($\chi^2$ p=$2.0 \times 10^{-71}$) with the 99 proteins higher in outer AF in the spatial proteome (*Figure 3I*). This indicates that key proteins defining a young outer AF is experiencing faster degradation in aged and degenerated samples.

Similarly, we identified two modules in the NP (*Figure 7H*), whereby one (magenta) corresponds to more degradation in the YND, and the other (blue) corresponds to more degradation in the AGD. Only 10 unique proteins were recorded in the magenta module (for YND), with COL2A1 being the most dominant (928 peptides); whereas 32 were recorded for the blue module (for AGD). Overall, there are more unique proteins involved in faster degradation in AGD AF and NP.

## MRI landscape correlates with proteomic landscape

We tested for a correlation between MRI signal intensity and proteome composition. In conventional 3T MRI of young discs, the NP is brightest reflecting its high hydration state while the AF is darker, thus less hydrated (*Figure 1B*; *Figure 8—figure supplement 1B*). Since aged discs present with more MRI phenotypes, we used higher resolution MRI (7T) on them (*Figure 1C*; *Figure 8A*), which showed less contrast between NP and AF than in the young discs. To enhance robustness, we obtained three transverse stacks per disc level for the aged discs (*Figure 8B*; *Figure 8—figure supplement 1D*), and averaged the pixel intensities for the different compartments showing that overall, the inner regions were still brighter than the outer (*Figure 8C*).

Next, we performed a level-compartment bi-clustering on the pixel intensities of the aged disc MRIs, which was bound by disc level and compartment (*Figure 8D*). The findings resembled the proteomic PCA (*Figure 2*) and clustering (*Figure 3—figure supplement 1V–W*) patterns. We performed a pixel intensity averaging of the disc compartments from the 3T images (*Figure 8—figure supplement 1D*), and a level-compartment bi-clustering on the pixel intensities (*Figure 8—figure supplement 1C*). While the clustering can clearly partition the inner from the outer disc compartments, the information value from each of the compartments is less due to the lower resolution of the MRI. In all, these results indicate a link between regional MRI landscapes and proteome profiles, prompting us to investigate their potential connections.

## Proteome-wide association with MRI landscapes reveals a hydration matrisome

The MRI and the static proteome were done on the same specimens in both individuals, so we could perform proteome-wide associations with the MRI intensities. We detected 85 significantly correlated ECM proteins, hereby referred to as the hydration matrisome (*Figure 8E*). We found no collagen to be positively correlated with brighter MRI, which fits current understanding as collagens contribute to fibrosis and dehydration. Other classes of matrisome proteins were either positively or negatively correlated, with differential components for each class (*Figure 8E*). Positively correlated proteoglycans included EPYC, PRG4 and VCAN, consistent with their normal expression in a young disc and hydration properties. Negatively correlated proteins included OAF (TNC, SLRPs) and fibrotic (POSTN) markers (*Figure 8E*).

Given this MRI-proteome link and the greater dynamic ranges of MRI in the aged discs enabled by the higher resolution 7T MRIs (*Figure 8D*), we hypothesised that the hydration matrisome might be used to provide information about MRI intensities and thus disc hydration. To test this, we trained a LASSO regression model (*Tibshirani, 1996*) of the aged MRIs using the hydration matrisome (85 proteins), and applied the model to predict the intensity of the MRIs of the young discs, based on the young proteome of the same 85 proteins. Remarkably, we obtained a PCC of 0.689 (p=$8.9 \times 10^{-6}$; Spearman = 0.776) between the actual and predicted MRI (*Figure 8—figure supplement 1E*). The predicted MRI intensities of the young disc exhibited a smooth monotonic decrease from the NP towards IAF, then dropped suddenly towards the OAF (*Figure 8F*, right panel), with an

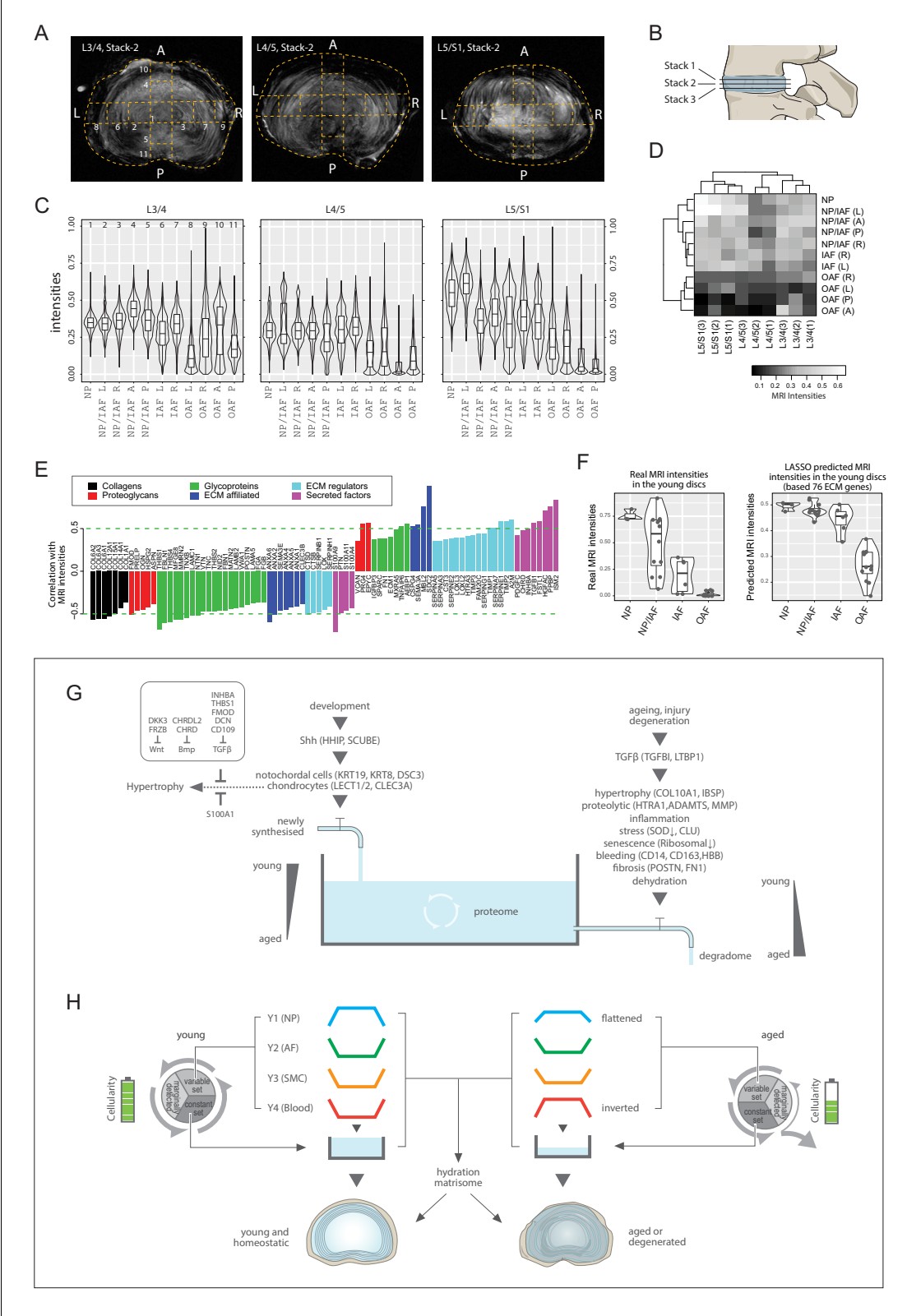

**Figure 8.** MRI intensities and their correlation with the proteomic data. (**A**) The middle MRI stack of each disc level in the aged cadaveric sample. (**B**) Schematic of the disc showing the three stacks of MRI images per disc. (**C**) Violin plots showing the pixel intensities within each location per disc level, corresponding to the respective locations taken for mass spectrometry measurements. Each violin-plot is the aggregate of three stacks of MRIs per disc. (**D**) A heatmap bi-clustering of levels and compartments based on the MRI intensities. (**E**) The hydration extracellular matrices (ECMs): the ECM

*Figure 8 continued on next page*

*Figure 8 continued*

proteins most positively and negatively correlated with MRI. (**F**) The 3T MRI intensities of the young discs across the compartments (left), and the predicted MRI intensities based on a LASSO regression model trained on the hydration ECMs (right). (**G**) A water-tank model of the dynamics in disc proteomics showing the balance of the proteome is maintained by adequate anabolism to balance catabolism. (**H**) Diagram showing the partitioning of the detected proteins into variable and constant sets, whereby four modules characterising the young healthy disc were further derived; and showing their changes with ageing. SMC: smooth muscle cell markers.

The online version of this article includes the following figure supplement(s) for figure 8:

**Figure supplement 1.** MRI-molecule connections.

ROC AUC (receiver operating characteristics, area under the curve) of 0.996 between IAF and OAF (*Figure 8—figure supplement 1F*). In comparison, actual MRIs exhibited a linear decrease from NP to OAF (*Figure 8F*, left panel). On reviewing these two patterns, we argue that the predicted intensities may be a more faithful representation of the young discs' water contents than the actual MRI, as it reflects the gross images (*Figure 1—figure supplement 1A*), PCA (*Figure 3A*) and Y1 modular trend in the young discs (*Figure 3E*). This exercise not only revealed the inherent connections between regional MRI and regional proteome, but also identified a set of ECM components that is predictive of MRI relating to disc hydration, which may be valuable for future clinical applications.

## Discussion

Here, we present DIPPER – a human IVD proteomic resource comprising point-reference genome-wide profiles. The discovery dataset was established from intact lumbar discs of a young cadaver with no history of skeletal abnormalities (e.g. scoliosis), and an aged cadaver with reduced IVD MRI intensity and annular tears. Although these two individuals may not be representative of their respective age-groups, this is the first known attempt to achieve high spatial resolution profiles in the discs, adding a critical and much needed dimension to the current available IVD proteomic datasets. We showed that our spatiotemporal proteomes integrate well with the dynamic proteome and transcriptome of clinical samples, demonstrating their application values with other datasets.

In creating the point-references, we use a well-established protein extraction protocol (*Önnerfjord et al., 2012*), and chromatographic fractionation of the peptides prior to mass spectrometry, we produced a dataset of the human IVD comprising 3,100 proteins, encompassing ~400 matrisome and 2,700 non-matrisome proteins, with 1,769 proteins detected in three or more profiles, considerably higher than recent studies (*Maseda et al., 2016*; *Rajasekaran et al., 2020*; *Ranjani et al., 2016*; *Sarath Babu et al., 2016*). The high quality of our data enabled the application of unbiased approaches including PCA and ANOVA to reveal the relative importance of the phenotypic factors. Particularly, age was found to be the dominant factor influencing proteome profiles.

Comparisons between different compartments of the young disc produced a reference landscape containing known (KRT8/19 for the NP; COL1A1, CILP and COMP for the AF) and novel (FRZB, CHRD, CHRDL2 for the NP; TNMD, SLRPs, and SOD1 for the AF) markers. The young healthy discs were enriched for matrisome components consistent with a healthy functional young IVD. Despite morphological differences between NP and IAF, the inner disc compartments (NP, NP/IAF, and IAF) display high similarities, in contrast to the large differences between IAF and OAF, which was consistent for discs from all lumbar disc levels. This morphological-molecular discrepancy might be accounted for by subtle differences in the ECM organisation, such as differences in GAG moieties on proteoglycans, or levels of glycosylation or other modifications of ECM that diversify function (*Silagi et al., 2018*). Nonetheless, we partitioned the detected proteins into a variable set that captures the diversity, and a constant set that lays the common foundation of all young compartments, which work in synergy to achieve disc function.

Clustering analysis of the 671 DEPs of the variable set identified four key modules (Y1-Y4). Visually, Y1 and Y2 mapped across the lateral and anteroposterior axes with opposing trends. Molecularly, module Y1 (NP) contained proteins promoting regulation of matrix remodelling, such as matrix degradation inhibitor MATN3 (*Jayasuriya et al., 2012*) and MMP inhibitor TIMP1. Inhibitors of WNT and BMP signalling were also present. Module Y2 (OAF) included COL1A1, THBS1/2/3, CILP1/CILP2, and TNMD, consistent with the OAF's tendon-like features (*Nakamichi et al., 2018*). It also included a set of SLRPs that might play roles in regulation of collagen assembly (*Robinson et al.,*

*2017*; *Taye et al., 2020*), fibril alignment (*Robinson et al., 2017*), maturation and crosslinking (*Kalamajski et al., 2016*), while others are known to inhibit or promote TGFβ signalling (*Markmann et al., 2000*). Notably, the composition of the IAF appeared to be a transition zone between NP and OAF rather than an independent compartment, as few proteins can distinguish it from adjacent compartments. Classes of proteins in both Y3 (smooth muscle feature) and Y4 (immune and blood) resemble Y2, which reflects the contractile property of the AF (*Nakai et al., 2016*) and the capillaries infiltrating or present at the superficial outer surface of the IVD.

In the aged disc, the DEPs between the inner and outer regions of the discs suggests extensive changes in the inner compartment(s). Mapping the aged data onto modules Y1-Y4 allowed a visualisation of the changes. The flattening of the Y1 and Y2 modules along both the lateral and antero-posterior axes indicated a convergence of the inner and outer disc. This is supported by the observed rapid decline of NP proteins and increase of AF proteins in the inner region. Fewer changes were seen in the aged OAF, which concurs with the notion that degenerative changes originate from the NP and radiate outwards; however, infringement of IAF into the NP cannot be excluded. The most marked change was seen in module Y4 (blood), where the pattern was inverted, characterised by high expression in the NP but low in OAF. While contamination cannot be excluded and there are reports that capillaries do not infiltrate the NP even in degeneration (*Nerlich et al., 2007*), our finding is consistent with other proteomic studies showing enrichment of blood proteins in pathological NP (*Maseda et al., 2016*). The route of infiltration can be from the fissured AF or cartilage endplates. Calcified endplates are more susceptible to microfractures, which can lead to blood infiltration into the NP (*Sun et al., 2020*). Of interest is the involvement of an immune response within the inner disc. This corroborates reports of inflammatory processes in ageing and degenerative discs, with the upregulation of pro-inflammatory cytokines and presence of inflammatory cells (*Molinos et al., 2015*; *Wuertz et al., 2012*).

The SILAC and degradome studies provided important insights into age-related differences in the biosynthetic and turnover activity in the IVD. The SILAC data indicated that protein synthesis is significantly impaired in aged degenerated discs. These findings correlate with reports of reduced cellularity in ageing (*Rodriguez et al., 2011*), which we have also ascertained by leveraging the relationship of histones and housekeeping genes with cell numbers. From the TAILS degradome analysis, we observed more cleaved protein fragments in aged compartments, particularly for structural proteins important for tissue integrity such as COMP and those involved in cell-matrix interactions such as FN1, which was coupled with the enrichment of the proteolytic process GO terms in the aged static proteome. Collectively, this reveals a systematic modification and replacement of the primary proteomic architecture of the young IVD with age that is associated with diminished or failure in functional properties in ageing or degeneration.

Despite known transcriptome-proteome discordance (*Fortelny et al., 2017*), our identification of their concordance allows insights into active changes in the young and aged discs. For example, inhibitors of the WNT pathway and antagonists of BMP/TGFβ signalling (*Leijten et al., 2013*) were down-regulated in the aged discs in both the proteome and transcriptome. Interestingly, the activation of these pathways is known to promote chondrocyte hypertrophy (*Dong et al., 2006*), and hypertrophy has been noted in IDD (*Rutges et al., 2010*). This suggests a model whereby the regulatory environment suppressing cellular hypertrophy changes with ageing or degeneration, resulting in conditions such as cellular senescence and tissue mineralisation that are part of the pathological process. In support, S100A1, a known inhibitor of chondrocyte hypertrophy (*Saito et al., 2007*) is downregulated, while chondrocyte hypertrophy markers *COL10A1* and *IBSP* (*Komori, 2010*) are upregulated in the aged disc. Similar changes have been observed in ageing mouse NP (*Veras et al., 2020*) as well as in osteoarthritis (*Zhu et al., 2009*) where chondrocyte hypertrophy is thought to be involved in its aetiology (*Ji et al., 2019*; *van der Kraan and van den Berg, 2012*). Given that WNT inhibitors are already in clinical trials for osteoarthritis (*Wang et al., 2019b*), this may point to a prospective therapeutic strategy for IDD.

A key finding of our study is the direct demonstration, within a single individual, of association between the hydration status of the disc as revealed by MRI, and the matrisome composition of the disc proteome. The remarkable correlation between predicted hydration states inferred from the spatial proteomic data and the high-definition phenotyping of the aged disc afforded by 7T MRI has enormous potential for understanding the molecular processes underlying IDD.

In conclusion, we have generated point-reference datasets of the young and aged disc proteome, at a significantly higher spatial resolution than previous works. By means of a methodological framework, we revealed compartmentalised information on the ECM composition and cellular activities, and their changes with ageing. Integration of this point-reference with additional age- and protein-specific information of synthesis/degradation helps gain insights into the underlying molecular pathology of degeneration (*Figure 8G and H*). The richness of information in DIPPER makes it a valuable resource for cross referencing with human, animal and *in vitro* studies to evaluate clinical relevance and guide the development of therapeutics for human IDD.

## Materials and methods

### Cadaveric specimens
Two human lumbar spines were obtained through approved regulations and governing bodies, with one young (16M) provided by L.H. (McGill University) and one aged (59M) from Articular Engineering, LLC (IL, USA). The young lumbar spine was received frozen as an intact whole lumbar spine. The aged lumbar spine was received frozen, dissected into bone-disc-bone segments. The cadaveric samples were stored at $-80^0$C until use.

### Clinical specimens
Clinical specimens were obtained with approval by the Institutional Review Board (references UW 13–576 and EC 1516–00 11/01/2001) and with informed consent in accordance with the Helsinki Declaration of 1975 (revision 1983) from another 15 patients undergoing surgery for IDD, trauma or adolescent idiopathic scoliosis at Queen Mary Hospital (Hong Kong), and Duchess of Kent Children's Hospital (Hong Kong). Information of both the cadaveric and clinical samples are summarised in *Table 1*.

### MRI imaging of cadaveric samples
The discs were thawed overnight at 4°C, and then pre-equalised for scanning at room temperature. For the young IVD, these were imaged together as the lumbar spine was kept intact. T2-weighted and T1-weighted sagittal and axial MRI, T1-rho MRI and Ultrashort-time-to-echo MRI images were obtained using a 3T Philips Achieva 3.0 system at the Department of Diagnostic Radiology, The University of Hong Kong.

For the aged discs, the IVD were imaged separately as bone-disc-bone segments, at the Department of Electrical and Electronic Engineering, The University of Hong Kong. The MRS and CEST imaging were performed. The FOV for the CEST imaging was adjusted to $76.8 \times 76.8$ mm$^2$ to accommodate the size of human lumbar discs (matrix size = $64 \times 64$, slice thickness = 2 mm). All MRI experiments were performed at room temperature using a 7 T pre-clinical scanner (70/16 Pharmascan, Bruker BioSpin GmbH, Germany) equipped with a 370 mT/m gradient system along each axis. Single-channel volume RF coils with different diameters were used for the samples based on size (60 mm for GAG phantoms and human cadaveric discs).

### Image assessment of the aged lumbar IVD
The MRI images in the transverse view were then assessed for intensity of the image (brighter signifying more water content). Three transverse MRI images per IVD were overlaid with a grid representing the areas that were cut for mass-spectrometry measurements as outlined previously. For each region, the 'intensity' was represented by the average of the pixel intensities, which were graphically visualised and used for correlative studies.

### Division of cadaveric IVD for mass spectrometry analysis
The endplates were carefully cut off with a scalpel, exposing the surface of the IVD, which were then cut into small segments spanning seven segments in the central left-right lateral axis, and five segments in the central anteroposterior axis (*Figure 1C*). In all, this corresponds to a total of 11 locations per IVD. Among them, four are from the OAF, two from the IAF (but only in the lateral axis), one from the central NP, and four from a transition zone between IAF and the NP (designated the 'NP/IAF'). Samples were stored frozen at $-80^0$C until use.

## SILAC by *ex vivo* culture of disc tissues

NP and AF disc tissues from spine surgeries (*Table 1*) were cultured in custom-made Arg- and Lys-free α-MEM (AthenaES) as per formulation of Gibco α-MEM (Cat #11900–024), supplemented with 10% dialysed FBS (10,000 MWCO, Biowest, Cat# S181D), penicillin/streptomycin, 2.2 g/L sodium bicarbonate (Sigma), 30 mg/L L-methionine (Sigma), 21 mg/L 'heavy' isotope-labelled $^{13}C_6$L-arginine (Arg6, Cambridge Isotopes, Cat # CLM-2265-H), 146 mg/L 'heavy' isotope-labelled 4,4,5,5-D4 L-Lysine (Lys4, Cambridge Isotopes, Cat # DLM-2640). Tissue explants were cultured for 7 days in hypoxia (1% $O_2$ and 5% $CO_2$ in air) at $37^0$C before being washed with PBS and frozen until use.

## Protein extraction and preparation for cadaveric and SILAC samples

The frozen samples were pulverised using a freezer mill (Spex) under liquid nitrogen. Samples were extracted using 15 volumes (w/v) of extraction buffer (4M guanidine hydrochloride (GuHCl), 50 mM sodium acetate, 100 mM 6-aminocaproic acid, and HALT protease inhibitor cocktail (Thermo Fischer Scientific), pH 5.8). Samples were mechanically dissociated with 10 freeze-thaw cycles and sonicated in a cold water bath, before extraction with gentle agitation at 4°C for 48 hr. Samples were centrifuged at 15,000 g for 30 min at 4°C and the supernatant was ethanol precipitated at a ratio of 1:9 for 16 hr at −20°C. The ethanol step was repeated and samples were centrifuged at 5000 g for 45 min at 4°C, and the protein pellets were air dried for 30 min.

Protein pellets were re-suspended in fresh 4M urea in 50 mM ammonium bicarbonate, pH 8, using water bath sonication to aid in the re-solubilisation of the samples. Samples underwent reduction with TCEP (5 mM final concentration) at 60°C for 1 hr, and alkylation with iodoacetamide (500 mM final concentration) for 20 min at RT. Protein concentration was measured using the BCA assay (Biorad) according to manufacturer's instructions. A total of 200 µg of protein was then buffer exchanged with 50 mM ammonium bicarbonate with centricon filters (Millipore, 30 kDa cutoff) according to manufacturer's instructions. Samples were digested with mass spec grade Trypsin/LysC (Promega) as per manufacturer's instructions. For SILAC-labelled samples, formic acid was added to a final concentration of 1%, and centrifuged and the supernatant then desalted prior to LC-MS/MS measurements. For the cadaveric samples, the digested peptides were then acidified with TFA (0.1% final concentration) and quantified using the peptide quantitative colorimetric peptide assay kit (Pierce, catalogue 23275) before undergoing fractionation using the High pH reversed phase peptide fractionation kit (Pierce, catalogue number 84868) into four fractions. Desalted peptides were dried, re-suspended in 0.1% formic acid prior to LC-MS/MS measurements.

## Mass spectrometry for cadaveric and SILAC samples

Samples were loaded onto the Dionex UltiMate 3000 RSLC nano Liquid Chromatography coupled to the Orbitrap Fusion Lumos Tribid Mass Spectrometer. Peptides were separated on a commercial Acclaim C18 column (75 µm internal diameter ×50 cm length, 1.9 µm particle size; Thermo). Separation was attained using a linear gradient of increasing buffer B (80% ACN and 0.1% formic acid) and declining buffer A (0.1% formic acid) at 300 nL/min. Buffer B was increased to 30% B in 210 min and ramped to 40% B in 10 min followed by a quick ramp to 95% B, where it was held for 5 min before a quick ramp back to 5% B, where it was held and the column was re-equilibrated. Mass spectrometer was operated in positive polarity mode with capillary temperature of 300°C. Full survey scan resolution was set to 120,000 with an automatic gain control (AGC) target value of $2 \times 10^6$, maximum ion injection time of 30 ms, and for a scan range of 400–1500 m/z. Data acquisition was in DDA mode to automatically isolate and fragment topN multiply charged precursors according to their intensities. Spectra were obtained at 30,000 MS2 resolution with AGC target of $1 \times 10^5$ and maximum ion injection time of 100 ms, 1.6 m/z isolation width, and normalised collisional energy of 31. Preceding precursor ions targeted for HCD were dynamically excluded of 50 s.

## Label-free quantitative data processing for cadaveric samples

Raw data were analysed using MaxQuant (v.1.6.3.3, Germany). Briefly, raw files were searched using Andromeda search engine against human UniProt protein database (20,395 entries, Oct 2018), supplemented with sequences of contaminant proteins. Andromeda search parameters for protein identification were set to a tolerance of 6 ppm for the parental peptide, and 20 ppm for fragmentation spectra and trypsin specificity allowing up to two miscleaved sites. Oxidation of methionine,

carboxyamidomethylation of cysteines was specified as a fixed modification. Minimal required peptide length was specified at seven amino acids. Peptides and proteins detected by at least two LFQ ion counts for each peptide in one of the samples were accepted, with a false discovery rate (FDR) of 1%. Proteins were quantified by normalised summed peptide intensities computed in MaxQuant with the LFQ option enabled. A total of 66 profiles were obtained: 11 locations $\times$ 3 disc levels $\times$ 2 individuals; with a median of 665 proteins (minimum 419, maximum 1920) per profile.

## Data processing for SILAC samples

The high-resolution, high mass accuracy mass spectrometry (MS) data obtained were processed using Proteome Discoverer (Ver 2.1), wherein data were searched using Sequest algorithm against Human UniProt database (29,900 entries, May 2016), supplemented with sequences of contaminant proteins, using the following search parameters settings: oxidised methionine (M), acetylation (Protein N-term), heavy Arginine (R6) and Lysine (K4) were selected as dynamic modifications, carboxyamidomethylation of cysteines was specified as a fixed modification, minimum peptide length of 7 amino acids was enabled, tolerance of 10 ppm for the parental peptide, and 20 ppm for fragmentation spectra, and trypsin specificity allowing up to two miscleaved sites. Confident proteins were identified using a target-decoy approach with a reversed database, strict FDR 1% at peptide and PSM level. Newly synthesised proteins were heavy labelled with Arg6- and Lys4, and the data was expressed as the normalised protein abundance obtained from heavy (labelled)/light (un-labelled) ratio.

## Degradome sample preparation, mass spectrometry, and data processing

Degradome analyses was performed on NP and AF from three non-degenerated and three degenerated individuals (*Table 1*). Frozen tissues were pulverised as described above and prepared for TAILS as previously reported (*Kleifeld et al., 2010*). After extraction with SDS buffer (1% SDS, 100 mM dithiothreitol, 1X protease inhibitor in deionised water) and sonication (three cycles, 15 s/cycle), the supernatant (soluble fraction) underwent reduction at 37°C and alkylation with a final concentration of 15 mM iodoacetamide for 30 min at RT. Samples were precipitated using chloroform/methanol, and the protein pellet air dried. Samples were re-suspended in 1M NaOH, quantified by nanodrop, diluted to 100 mM HEPES and 4M GnHCl and pH adjusted (pH6.5–7.5) prior to 6-plex TMT labelling as per manufacturer's instructions (Sixplex TMT, Cat# 90061, ThermoFisher Scientific). Equal ratios of TMT-labelled samples were pooled and methanol/chloroform precipitated. Protein pellets were air-dried and re-suspended in 200 mM HEPES (pH8), and digested with trypsin (1:100 ratio) for 16 hr at 37°C, pH 6.5 and a sample was taken for pre-TAILS. High-molecular-weight dendritic polyglycerol aldehyde polymer (ratio of 5:1 w/w polymer to sample) and NaBH$_3$CN (to a final concentration of 80 mM) was added, incubated at 37°C for 16 hr, followed by quenching with 100 mM ethanolamine (30 min at 37°C) and underwent ultrafiltration (MWCO of 10,000). Collected samples were desalted, acidified to 0.1% formic acid and dried, prior to MS analysis.

Samples were analysed on a Thermo Scientific Easy nLC-1000 coupled online to a Bruker Daltonics Impact II UHR QTOF. Briefly, peptides were loaded onto a 20 cm x 75 μm I.D. analytical column packed with 1.8 μm C18 material (Dr. Maisch GmbH, Germany) in 100% buffer A (99.9% H$_2$O, 0.1% formic acid) at 800 bar followed by a linear gradient elution in buffer B (99.9% acetonitrile, 0.1% formic acid) to a final 30% buffer B for a total 180 min including washing with 95% buffer B. Eluted peptides were ionised by ESI and peptide ions were subjected to tandem MS analysis using a data-dependent acquisition method. A top17 method was employed, where the top 17 most intense multiply charged precursor ions were isolated for MS/MS using collision-induced-dissociation, and actively excluded for 30 s.

MGF files were extracted and searched using Mascot against the UniProt *Homo sapiens* database, with semi-ArgC specificity, TMT6plex quantification, variable oxidation of methionine, variable acetylation of N termini, 20 ppm MS1 error tolerance, 0.05 Da MS2 error tolerance and two missed cleavages. Mascot. dat files were imported into Scaffold Q+S v4.4.3 for peptide identification processing to a final FDR of 1%. Quantitative values were calculated through Scaffold and used for subsequent analyses.

## Transcriptomic samples: isolation, RNA extraction, and data processing

AF and NP tissues from four individuals were cut into approximately 0.5 cm$^3$ pieces, and put into the Dulbecco's modified Eagle's medium (DMEM) (Gibco) supplemented with 20 mM HEPES (USB), 1% penicillin-streptomycin (Gibco) and 0.4% fungizone (Gibco). The tissues were digested with 0.2% pronase (Roche) for 1 hr, and centrifuged at 200 g for 5 min to remove supernatant. AF and NP were then digested by 0.1% type II collagenase (Worthington Biochemical) for 14 hr and 0.05% type II collagenase for 8 hr, respectively. Cell suspension was filtered through a 70 µm cell strainer (BD Falcon) and centrifuged at 200 g for 5 min. The cell pellet was washed with phosphate buffered saline (PBS) and centrifuged again to remove the supernatant. RNA was then extracted from the isolated disc cells using Absolutely RNA Nanoprep Kit (Stratagene), following manufacturer's protocol, and stored at −80°C until further processing.

The quality and quantity of total RNA were assessed on the Bioanalyzer (Agilent) using the RNA 6000 Nano total RNA assay. cDNA was generated using Affymetrix GeneChip Two-Cycle cDNA Synthesis Kit, followed by *in vitro* transcription to produce biotin-labelled cRNA. The sample was then hybridised onto the Affymetrix GeneChip Human Genome U133 Plus 2.0 Array. The array image, CEL file and other related files were generated using Affymetrix GeneChip Command Console. The experiment was conducted as a service at the Centre for PanorOmic Sciences of the University of Hong Kong.

CEL and other files were loaded into GeneSpring GX 10 (Agilent) software. The RMA algorithm was used for probe summation. Data were normalised with baseline transformed to median of all samples. A loose filtering based on the raw intensity values was then applied to remove background noise. Consequently, transcriptomic data with a total of 54,675 probes (corresponding to 20,887 genes) and eight profiles were obtained.

## Bioinformatics and functional analyses

The detected proteins were compared against the transcription factor (TF) database (*Vaquerizas et al., 2009*) and the human genome nomenclature consortium database for cell surface markers (CDs) (*Braschi et al., 2019*), where 77 TFs and 83 CDs were detected (*Figure 1—figure supplement 3A*). Excluding missing values, the LFQ levels among the data-points range from 15.6 to 41.1, with a Gaussian-like empirical distribution (*Figure 2—figure supplement 1A*). The numbers of valid values per protein were found to decline rapidly when they were sorted in descending order (*Figure 2—figure supplement 1B*, upper panel). To perform PCAs, only a subset of genes with sufficiently large numbers of valid values (i.e. non-missing values) were used. The cutoff for this was chosen based on a point corresponding to the steepest slope of descending order of valid protein numbers (*Figure 2—figure supplement 1B*, second panel), such that the increase of valid values is slower than the increase of missing values beyond that point. Subsequently, the top 507 genes were picked representing 59.8% of all valid values. This new subset includes 12.4% of all missing values. Since the subset of data still contains some missing values, an imputation strategy was adopted employing the Multiple Imputation by Chained Equations (MICE) method and package (*Buuren and Groothuis-Oudshoorn, 2011*), with a max iteration set at 50 and the default PMM method (predictive mean matching). To further ensure normality, Winsorisation was applied such that genes whose average is below 5% or above 95% of all genes were also excluded from PCA. The data was then profile-wise standardised (zero-mean and one standard deviation) before PCA was applied on the R platform (*R Development Core Team, 2013*).

To assess the impact of the spatiotemporal factors on the proteomic profiles, we performed Analysis of Variance (ANOVA), correlating each protein to the age, compartments, level, and directionality. To draw the soft boundaries on the PCA plot between groups of samples, support vector machines with polynomial (degree of 2) kernel were applied using the LIBSVM package (*Chang and Lin, 2011*) and the PCA coordinates as inputs for training. A meshed grid covering the whole PCA field was created to make prediction and draw probability contours for −0.5, 0, and 0.5 from the fitted model. Hierarchical clustering was performed with (1- correlation coefficient) as the distance metrics unless otherwise specified.

To address the problem of 'dropout' effects while avoiding extra inter-dependency introduced due to imputations, we adopted three strategies in calculating the DEPs, namely, by statistical testing, exclusively detected, and fold-change cutoff approaches. First, for the proteins that have over

half valid values in both groups under comparison, we performed t-testing with p-values adjusted for multiple testing by the false discovery rate (FDR). Those with FDR below 0.05 were considered statistical DEPs. Second, for the proteins where one group has some valid values while the other group is completely not detected, we considered the ones with over half valid values in one group to be exclusive DEPs. For those proteins that were expressed in <50% in both groups, the ones with fold-change greater than two were also considered to be DEPs.

To fit the lateral and anteroposterior trends for the modules of genes identified in the young samples, a Gaussian Process Estimation (GPE) model was trained using the GauPro package in *R Development Core Team, 2013*. Pathway analyses was conducted on the GSEA (*Subramanian et al., 2005*). Signalling proteins was compiled based on 25 Signal transduction pathways listed on KEGG (*Kanehisa et al., 2019*).

For transcriptomic data, we used a thresholding approach to detect DEGs (differentially expressed genes), whereby a gene was considered a DEG if the $\log_2$(fold-change) is greater than three and the average expression (logarithmic scale) is greater than 10 (*Figure 6—figure supplement 1E–H*).

The LASSO model between MRI and proteome was trained using the R package 'glmnet', wherein the 85 ECMs were first imputed for missing values in them using MICE. Nine ECMs were not imputed for too many missing values, leaving 76 for training and testing. The best value for λ was determined by cross-validations. A model was then trained on the aged MRIs (dependent variable) and aged proteome of the 76 genes (independent variable). The fitted model was then applied to the young proteome to predict MRIs of the young discs.

## Software availability

The custom scripts for processing and analysing the data were housed at github.com/hkudclab/DIP-PER; *Tam, 2021*; copy archived at swh:1:rev:49e4da79786de1dfa496d7c7472343f3b865696c. An interactive web interface for the data is available at http://www.sbms.hku.hk/dclab/DIPPER/.

## Acknowledgements

We thank Dr Dino Samartzis for arranging the MRI of the young lumbar spine, and Prof. Kenneth Cheung and Dr Jason Cheung for collecting surgical disc specimens. We thank Dr Ed Wu and Dr Anna Wang of the Department of Electrical and Electronic Engineering at HKU for performing the high-resolution MRI on the aged discs. Part of this work was supported by the Theme-based Research Scheme (T12-708/12N) and Area of Excellence (AoE/M-04/04) of the Hong Kong Research Grants Council (RGC) (Kathryn Cheah, Danny Chan), the RGC European Union - Hong Kong Research and Innovation Cooperation Co-funding Mechanism (iPSpine) (E-HKU703/18) (Danny Chan), and by the Ministry of Science and Technology of the People's Republic of China: National Strategic Basic Research Program ('973') (2014CB942900) (Danny Chan). The TAILS analyses were supported by a Canadian Institutes of Health Research210312 Foundation Grant (FDN-148408) (Christopher Overall).

## Additional information

### Competing interests

Kathryn Song Eng Cheah: Senior editor, *eLife*. Theo Klein: Theo Klein is affiliated with Triskelion BV. The author has no financial interests to declare. The other authors declare that no competing interests exist.

### Funding

| Funder | Grant reference number | Author |
| --- | --- | --- |
| Research Grants Council, University Grants Committee | E-HKU703/18 | Danny Chan |
| Research Grants Council, University Grants Committee | T12-708/12N | Kathryn Song Eng Cheah |

| Research Grants Council, University Grants Committee | AoE/M-04/04 | Kathryn Song Eng Cheah |
| Ministry of Science and Technology of the People's Republic of China | ("973") (2014CB942900) | Danny Chan |
| Canadian Institutes of Health Research | FDN-148408 | Christopher M Overall |

The funders had no role in study design, data collection and interpretation, or the decision to submit the work for publication.

## Author contributions

Vivian Tam, Conceptualization, Data curation, Formal analysis, Validation, Investigation, Visualization, Methodology, Writing - original draft, Writing - review and editing; Peikai Chen, Conceptualization, Data curation, Software, Formal analysis, Investigation, Visualization, Methodology, Writing - original draft, Writing - review and editing; Anita Yee, Data curation, Formal analysis, Methodology, Writing - review and editing; Nestor Solis, Data curation, Methodology, Writing - review and editing; Theo Klein, Rakesh Sharma, Data curation, Writing - review and editing; Mateusz Kudelko, Data curation, Formal analysis, Writing - review and editing; Wilson CW Chan, Formal analysis, Writing - review and editing; Christopher M Overall, Conceptualization, Supervision, Writing - review and editing; Lisbet Haglund, Conceptualization, Resources, Writing - review and editing; Pak C Sham, Resources, Methodology, Writing - review and editing; Kathryn Song Eng Cheah, Conceptualization, Resources, Supervision, Funding acquisition, Writing - review and editing; Danny Chan, Conceptualization, Resources, Formal analysis, Supervision, Funding acquisition, Visualization, Methodology, Writing - original draft, Project administration, Writing - review and editing

## Author ORCIDs

Vivian Tam (iD) https://orcid.org/0000-0002-0457-3477
Peikai Chen (iD) https://orcid.org/0000-0003-1880-0893
Theo Klein (iD) http://orcid.org/0000-0001-8061-9353
Lisbet Haglund (iD) http://orcid.org/0000-0002-1288-2149
Kathryn Song Eng Cheah (iD) http://orcid.org/0000-0003-0802-8799
Danny Chan (iD) https://orcid.org/0000-0003-3824-5778

## Ethics

Human subjects: Clinical specimens were obtained with approval by the Institutional Review Board (references UW13-576 and EC 1516-00 11/01/2001) and with informed consent in accordance with the Helsinki Declaration of 1975 (revision 1983).

## Decision letter and Author response

Decision letter https://doi.org/10.7554/eLife.64940.sa1
Author response https://doi.org/10.7554/eLife.64940.sa2

## Additional files

### Supplementary files

• Supplementary file 1. Processed data of the 66 LC-MS/MS static spatial proteome profiles.

• Supplementary file 2. Differentially expressed proteins (DEPs) among pairs of sample groups within the 33 young static spatial disc profiles.

• Supplementary file 3. Differentially expressed proteins (DEPs) among pairs of sample groups within the 33 aged static spatial disc profiles.

• Supplementary file 4. Differentially expressed proteins (DEPs) between young and aged sample groups of static spatial proteomes.

- Supplementary file 5. Significantly enriched gene ontology (GO) terms associated with proteins expressed higher in all young or all aged discs.
- Transparent reporting form

## Data availability

The mass spectrometry proteomics raw data have been deposited to the ProteomeXchange Consortium via the PRIDE repository with the following dataset identifiers for cadaver samples (PXD017740), SILAC samples (PXD018193), and degradome samples (PXD018298). The RAW data for the transcriptome data has been deposited on NCBI GEO with accession number GSE147383.

The following datasets were generated:

| Author(s) | Year | Dataset title | Dataset URL | Database and Identifier |
|---|---|---|---|---|
| Tam V, Chan D | 2020 | The degradome of the human intervertebral disc | http://www.ebi.ac.uk/pride/archive/projects/PXD018298 | PRIDE, PXD018298 |
| Tam V, Chan D | 2020 | A proteomic architectural landscape of the healthy and aging human intervertebral disc | http://www.ebi.ac.uk/pride/archive/projects/PXD017740 | PRIDE, PXD017740 |
| Tam V, Chan D | 2020 | Actively synthesised proteins in human intervertebral disc | http://www.ebi.ac.uk/pride/archive/projects/PXD018193 | PRIDE, PXD018193 |
| Yee A, Tam V, Chen P, Chan D | 2020 | Gene expression data for human intervertebral discs | https://www.ncbi.nlm.nih.gov/geo/query/acc.cgi?acc=GSE147383 | NCBI Gene Expression Omnibus, GSE147383 |

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
