## [Decision Letter]

**Acceptance summary:**

Inter vertebral disc (IVD) degeneration is associated with low back pain and significantly affects patient's quality of life. The pathophysiology of IVD disease is poorly understood and currently no treatments are available to restore disc structure and function. The spatial-temporal proteomic and transcriptomic data presented in this manuscript on lumbar discs of healthy young and old cadavers provide an important resource for researchers in this field that will facilitate future identification of therapeutic targets for this debilitating disease.

**Decision letter after peer review:**

[Editors’ note: the authors submitted for reconsideration following the decision after peer review. What follows is the decision letter after the first round of review.]

Thank you for submitting your work entitled "The spatial proteome of the human intervertebral disc reveals architectural changes in health, ageing and degeneration" for consideration by *eLife*. Your article has been reviewed by a Senior Editor, a Reviewing Editor, and three reviewers. The reviewers have opted to remain anonymous.

Our decision has been reached after consultation between the reviewers. Based on these discussions and the individual reviews below, we regret to inform you that your work will not be considered further for publication in *eLife*.

While all three reviewers complemented on the comprehensive computational analyses of the proteomics data, there was a major concern that all of the analyses were done using data derived from a single young and old IVD disc. Furthermore, the IVD from a scoliosis patient may not be a good control to represent normal young IVD because of the known effect of scoliosis on cell behavior and gene expression. In general, the study is considered descriptive and correlative in nature.

Reviewer #1:

This paper explores at the proteomic and transcriptomic level, the profile shifts that occur with ageing/degeneration particularly focusing on different intervertebral disc compartments and on distinct segmental levels.

It is a new and extremely detailed study that provides an immense amount of data on intervertebral disc spatial proteomics which is foreseen to have an important contribute for the field with an expected impact in the development of novel therapeutic alternatives.

Despite the exhaustive analysis of the obtained data (from different levels and areas of the disc), for most of the comparisons, conclusions are inferred from a single individual per group, with its own genetic background and biological particularities. My major concern regards the fact that many extrapolations might differ in samples from other individuals. In Figure 7B, for instance, simply by comparing 2 individuals it is clear that the profile of NP_YND151 is very different from that of NP_YND152, as stated by the authors, which corroborates this issue of biological variability and how it may impact on the conclusions taken.

Reviewer #2:

This manuscript using dynamic proteomic analysis of SILAC and TAILS degradome, established the reference spatial proteome of young healthy human IVDs by profiling the 11 different regions of disc. Additionally, aging associated changes in IVD proteomic landscape was reported with emphasis on core and core-associated matrisome. Using series of valid computational analyses, authors have evaluated the similarities and differences in the young and aged disc compartments. This will be a valuable resource in IDD research specifically by providing an interactive web interface to comprehend the proteome data. Study design is good and logical and experiments were well performed and organized. Conclusion drawn from experiment was convincing with data presented. There are few concerns for the author that need to be addressed:

It is not clear why author chose to use only one lumber disc to establish the reference proteome, when there is availability of 17 individual lumber discs. Adding more samples in establishing the new reference proteome would certainly add rigor to data.

It would be interesting to see the “LASSO regression model” to provide association between MRI and disc hydration is also valid in young disc. Is this regression specific to degenerated aged disc?

How did author ensure the validity of imputation analysis for the missing data? What was the iteration value used? How was the data normalized?

Did author identify any transcription factor associated module in expression module Y analysis of young and aged disc. Since transcription factors define the cellular identity, it would be interesting to see the compartments specific the key transcript factors in these IVDs.

In DEG analysis, why did author see only 88 gene between young and aged disc, considering huge transcriptomic changes during disc degeneration, this number is relatively low. What was significant cutoff used for DEG? Was multiple correction testing applied?

Reviewer #3:

This work employs a sophisticated proteomic analysis with SILAC and TAILS methods to provide changes in proteomic landscape in young and aging human disc. Technical aspects of the study are interesting although the information generated is more or less been known for a while including a proteomic study done by this group in 2016. While 33 samples were analyzed each for young and old discs, they all are derived from a single donor. It is known that there are individual to individual changes which are therefore not accounted for in proteomic analysis. One of the major issues in this study is the use of scoliotic tissue specimens used for transcriptomic, SILAC and TAILS analyses. It is abundantly clear that scoliosis and other spinal deformities alter cell behavior and gene expression profiles due to long term altered mechanical loading on discs and while MRI analysis could show "white" discs, they cannot be regarded normal young discs. Consequently, the profiles and information that is generated with such tissues also needs to carefully interpreted. Study is also descriptive in nature and mostly correlative. So, while methodologically interesting analyses is presented the amount of new biological insights generated are limited, most confirmatory to what is largely known e.g. fibrotic changes of NP and acquisition of AF like profiles, changes in signaling pathways SHH, WNT and TGF. I do not think this manuscript warrants publication in *eLife*.

---

## [Author Response]

While all three reviewers complemented on the comprehensive computational analyses of the proteomics data, there was a major concern that all of the analyses were done using data derived from a single young and old IVD disc. Furthermore, the IVD from a scoliosis patient may not be a good control to represent normal young IVD because of the known effect of scoliosis on cell behavior and gene expression. In general, the study is considered descriptive and correlative in nature.Reviewer #1:1) This paper explores at the proteomic and transcriptomic level, the profile shifts that occur with ageing/degeneration particularly focusing on different intervertebral disc compartments and on distinct segmental levels.It is a new and extremely detailed study that provides an immense amount of data on intervertebral disc spatial proteomics which is foreseen to have an important contribute for the field with an expected impact in the development of novel therapeutic alternatives.Despite the exhaustive analysis of the obtained data (from different levels and areas of the disc), for most of the comparisons, conclusions are inferred from a single individual per group, with its own genetic background and biological particularities. My major concern regards the fact that many extrapolations might differ in samples from other individuals. In Figure 7B, for instance, simply by comparing 2 individuals it is clear that the profile of NP_YND151 is very different from that of NP_YND152, as stated by the authors, which corroborates this issue of biological variability and how it may impact on the conclusions taken.

We thank the reviewer for their comments, particularly for the affirmative comments on the overall design, methodology and potential contribution to the field.

With regards to the comment on the single individual per group in our static proteome data, we agree that it is the limitation of the design. However, we would like to clarify a few points. First, the young individual is a trauma-induced sample (not reported to have degeneration or scoliosis) and the aged sample is not known to have a diagnosis of degeneration, although its MRI does show features typical of his age (Figure 1C). These samples are incredibly rare to obtain, and it is even rarer to obtain multiple intact discs from the same individual (Figure 1BC). Second, we obtained multiple disc levels and directions, amounting to 33 profiles per individual (Figure 1D). These 33 profiles are not simply just technical replicates of each other, but are separate measurements for samples with different location information (levels, directions, compartment). Taking the analogy of single cell transcriptome analysis, one can imagine that this study design lies halfway between bulk and single-cell proteomics, but with additional locational information. Thus, they present much more information than a single bulk profile per individual. Third, we also have another 15 individuals (clinical samples) for the dynamic proteome (SILAC and degradome) and transcriptome (Figure 1A). Despite slight clinical variations and different measuring variables or quantities, they demonstrate consistent behaviour with the static proteome (Figure 6 and Figure 7), which adds a further dimension of robustness to the findings in the static proteome. Finally, in recognising the sample size limitation, we do not claim these samples are representative of the general population, but it is hoped that the data are highly valuable as point references for the disc proteome. This is clarified in the text in the second paragraph of the Discussion.

The lower number of proteins in the heavy SILAC profile of NP_YND152 is likely a result of a technical issue during sample preparation, evidenced by the less dramatic reduction in its light SILAC profile (middle panel, Figure 7B). Despite its lower quality, NP_YND152 still behaves biologically more similar to NP_YND151 than to both AF_YND and AGD (AF and NP) samples (as illustrated in Figure 7—figure supplement 1C). Thus, it is still representative of a young disc. This is the only technically compromised sample (Figure 7B) whereas the AF_YND samples are more technically stable with lower observed variance (Figure 7B). We have clarified this in the text (subsection “Aged or degenerated discs synthesise fewer proteins”).

Reviewer #2:1) This manuscript using dynamic proteomic analysis of SILAC and TAILS degradome, established the reference spatial proteome of young healthy human IVDs by profiling the 11 different regions of disc. Additionally, aging associated changes in IVD proteomic landscape was reported with emphasis on core and core-associated matrisome. Using series of valid computational analyses, authors have evaluated the similarities and differences in the young and aged disc compartments. This will be a valuable resource in IDD research specifically by providing an interactive web interface to comprehend the proteome data. Study design is good and logical and experiments were well performed and organized. Conclusion drawn from experiment was convincing with data presented. There are few concerns for the author that need to be addressed:

We thank the reviewer for their positive comments on our approaches and analytical framework, and for agreeing that this is a valuable resource in IVD research. In light of this, we are presenting the current version as a resource paper.

2) It is not clear why author chose to use only one lumber disc to establish the reference proteome, when there is availability of 17 individual lumber discs. Adding more samples in establishing the new reference proteome would certainly add rigor to data.

We thank the reviewer for their comment on the sample size issue. We would like to clarify a few points. First, the 17 individuals actually comprise two groups (Table 1 and Materials and methods), one that was from cadaveric samples (n=2; not reported to have scoliosis or degeneration, now clarified in our current texts at multiple locations, including the Introduction, Results, and Discussion), and the other that consists of clinical samples (n=15; young samples are scoliotic, and aged samples mostly have degeneration). For the cadaveric samples, multiple intact levels can be available up to our design and choice, whereas the disc levels of the clinical samples available to us depend on the surgery performed on each individual. As such, the clinical samples are usually from one level, which differed from patient to patient. They also tend to not be a whole intact disc. Therefore, it is neither highly valuable nor suitable to perform spatial proteome on them.

In practice, we profiled 11 localities per disc level for each of the respective three lumbar levels, for both cadaveric spines (Figure 1D). This amounts to 66 static spatial proteome profiles, wherein neither scoliosis nor degeneration is a known factor. To our knowledge, this is the first attempt at profiling the disc proteome at such high spatial resolution. The levels, directions and compartments are directly matched between the two cadavers, which makes the comparisons highly meaningful.

We are fully aware and agree that having one subject per age-group in our static proteome is not ideal to extrapolate situations in the general population. However, obtaining a sizeable number of intact cadaveric spines per age-group may be dauntingly challenging. We argue that in spite of the potential under-representation of these two cadaveric spines, the multi-level, multi-locality proteomic profiles thus obtained and the derived results, can still serve as critical point references for gaining insights on spatial changes occurring during normal ageing, especially as no such data is known to exist.

Additionally, the clinical samples (n=15) afforded us to probe other aspects of molecular dynamics (protein turnover and transcriptome) for samples that are directly relevant to disc degeneration. The concordant results among these integrative multi-omics data (Figure 6 and Figure 7) also paint a coherent picture of the pathobiology under disc homeostasis and ageing/degeneration.

We added some explanation relating to these in subsection “Disc samples and their phenotypes”.

3) It would be interesting to see the “LASSO regression model” to provide association between MRI and disc hydration is also valid in young disc. Is this regression specific to degenerated aged disc?

We thank the reviewer for their interest in the LASSO model. In short, we applied this only to the aged sample. The reasons are two-fold. First, the aged samples are naturally and biologically more variable. Second, the MRIs for aged samples have much higher resolution at 7T. These provide a larger dynamic range to afford a regression model to be practical. On the contrary, the young discs show a lower variability in both proteome and MRI intensity (3T), and thus may not afford sufficient dynamic ranges to perform such a regression.

4) How did author ensure the validity of imputation analysis for the missing data? What was the iteration value used? How was the data normalized?

We thank the reviewer for the interest in the technical aspect on data imputation. We used the “mice” package of R for performing imputation on a set of 507 genes selected based on an approach illustrated in Figure 2—figure supplement 1A-C. The parameter for the imputation is: [m=5,maxit=50,meth=“pmm”,seed=500], where m is the number of multiple imputations, “maxit” is the maximum number of iterations, “meth” is the imputation method, and “seed” is the seeding for random number. This was indeed mentioned specifically in subsection “Bioinformatics” and can also be checked on our scripts publicly available on github: https://github.com/hkudclab/DIPPER/blob/master/Figure2/PCA.R.

5) Did author identify any transcription factor associated module in expression module Y analysis of young and aged disc. Since transcription factors define the cellular identity, it would be interesting to see the compartments specific the key transcript factors in these IVDs.

We thank the reviewer for their comment. We truly agree that transcription factors play a critical role in deciphering the regulatory mechanisms underlying cellular and tissue functions. In fact, we had a whole sub-section dedicated to reviewing the transcription factors and other key cellular proteins in the static proteome (subsection “Cellular activities inferred from non-matrisome proteins”).

In short, due to the nature of proteomic measurements, the bulk of materials came from the extracellular matrix, with much smaller contribution coming from intracellular proteins. This is especially relevant here as the IVD is much more enriched for matrix than for cells. Additionally, due to their regulatory roles, it is known that only small amounts of transcription factors are needed, making them even less detectable as compared with other intracellular proteins.

Nonetheless, we still detected a small number (82/3,100; Figure 1—figure supplement 3A) of transcription factors or DNA binding proteins. But many of these are histones, which leaves us with very few regulatory transcription factors.

We reviewed the genes in the four Y modules (Figure 3—figure supplement 3) and found that only two DNA binding proteins (MECP2 and HIST1H1B) were present in Y1 and Y2 modules, which we have now highlighted red, but otherwise there is no classical transcription factors in these modules. This is in contrast to the many well-known notochordal or NP markers, or AF markers that were in them.

Notwithstanding, we did detect some transcription factors through our transcriptome data, which measured mostly cellular materials. For example, we detected ZNF207, ZNF638, and HOPX to be down-regulated in aged or degenerated NP (versus young NP) (Figure 6B), whereby HOPX is reported to be differentially expressed in mouse NP as compared to AF (Veras et al., 2020), and reported to be expressed in mouse notochordal NP cells (Lam, 2013).

6) In DEG analysis, why did author see only 88 gene between young and aged disc, considering huge transcriptomic changes during disc degeneration, this number is relatively low. What was significant cutoff used for DEG? Was multiple correction testing applied?

We thank the reviewer for their comment on DEG results of subsection “Transcriptome shows AF-like characteristics of the aged/degenerated NP” and Figure 6—figure supplement 1E-H.

We would like to clarify that the 88 DEGs were not reported as comparison results between young and aged discs, but are that between young NP and young AF (Figure 6—figure supplement 1E). 39 of these 88 are expressed higher in young NP and 49 are higher in young AF.

For the comparisons between young and aged NP, there are 216 DEGs (Figure 6—figure supplement 1H), which is considerably more than 88, and fits what the reviewer says: “huge transcriptomic changes during disc degeneration”. There is also an elaboration on the larger number of AF-vsNP DEGs in aged samples than in young samples in subsection “Transcriptome shows AF-like characteristics of the aged/degenerated NP”.

Since there are only two samples per group in each of the comparisons in Figure 6—figure supplement 1E-H, we used a thresholding approach to detect DEGs, whereby a gene was considered a DEG if the log2(fold-change) is greater than 3 and the average expression (log scale) is greater than 10 (Materials and methods).

Reviewer #3:1) This work employs a sophisticated proteomic analysis with SILAC and TAILS methods to provide changes in proteomic landscape in young and aging human disc.

We thank the reviewer for appreciating the intricate designs in our approaches. The level of sophistication was needed to integrate multiple data types, and to de-convolute the effects of multiple factors on the disc proteome.

2) Technical aspects of the study are interesting although the information generated is more or less been known for a while including a proteomic study done by this group in 2016.

We appreciate the reviewer’s interest in the technical aspects of our study, and recognition of our group’s previous work in 2016.

However, we believe this study adds a much higher level of insights that were not able to be addressed in our previous 2016 paper (Yee et al., 2016), and other relevant papers (Babu et al. 2016; Rajasekaran et al., 2020). For example, we provide spatial information on the static proteome that covers levels, directions and compartments in aged and young disc (Figure 1A) that have not been addressed previously in any single publication.

Furthermore, considering the static proteome is an accumulation of the life-time of the protein turnover, the amounts of synthesis and turnover are likely to be different in young and aged discs, thus we also addressed this by the dynamic proteome (synthesis of new proteins – SILAC, and degradome, Figure 7).

Our dataset also included transcriptome of young and aged disc (Figure 6) which complemented our proteomic data to give further depth into the understanding of the cell states, as the proteome mostly measure the extracellular materials where the transcriptome measures the cellular materials.

From our data and high-resolution MRI images of aged disc, we were also able to identify a “hydration matrisome” and modelled this back to young MRI (Figure 8), which has the potential to serve as diagnostic markers for clinical application. It is rare that multiple aspects (static, dynamic proteome, transcriptome, MRI) can be unified within a single study. Although some of these findings were previously reported, they were disconnected or weakly implicated. Whereas in our study, multiple datasets and integrative analyses point in the same directions, which reinforced the robustness of the findings.

3) While 33 samples were analyzed each for young and old discs, they all are derived from a single donor. It is known that there are individual to individual changes which are therefore not accounted for in proteomic analysis.

We thank the reviewer for their comment and we understand the concern. We would like to refer you to our response to reviewer 1, question 1 and our response to reviewer 2 question 2 for a thorough explanation.

4) One of the major issues in this study is the use of scoliotic tissue specimens used for transcriptomic, SILAC and TAILS analyses. It is abundantly clear that scoliosis and other spinal deformities alter cell behavior and gene expression profiles due to long term altered mechanical loading on discs and while MRI analysis could show "white" discs, they cannot be regarded normal young discs. Consequently, the profiles and information that is generated with such tissues also needs to carefully interpreted.

We thank the reviewer for their comments and we understand the concern. We agree that (except in static proteome) the young non-degenerated samples used in transcriptomics, SILAC and TAILS are indeed scoliotic. We are aware there are recent reports on comparing non-degenerated and scoliotic NP with indication of some molecular differences (PMID: 32945197).

However, we would like to emphasise that the static spatial proteome (used as a “discovery set” here) was indeed performed on young disc that had no reported scoliotic diagnosis or degeneration, which can also be confirmed by MRI (Figure 1B). We have also clarified this in our texts at multiple locations, including the Introduction, Discussion, and subsection “Disc samples and their phenotypes”.

We agree with the reviewer that scoliosis is not “normal”. In fact, we never claimed that the scoliotic samples used in our non-spatial data were “normal”. Instead, we referred to them as “young non-degenerated (YND)”.

Actually, results of non-spatial proteome data were primarily for cross-comparison and validation purposes (Figure 1A).

Strikingly, despite the technological differences between microarray and proteomics and known RNA-protein discrepancy, the results yielded by cross-age comparisons in the static proteome (where scoliotic samples were not involved) was mutually supportive with that yielded by the transcriptome (Figure 6). For example, the notochordal markers KRT8, KRT19 and CHRD (Figure 6C) are consistently expressed at higher levels in both proteome and transcriptome of young NP, with or without scoliosis. Likewise, in outer AF, markers such as COL1A1, THBS1/2/4, and COL12A1 is consistently expressed higher in both transcriptome and proteome.

5) Study is also descriptive in nature and mostly correlative. So, while methodologically interesting analyses is presented the amount of new biological insights generated are limited, most confirmatory to what is largely known e.g. fibrotic changes of NP and acquisition of AF like profiles, changes in signaling pathways SHH, WNT and TGF. I do not think this manuscript warrants publication in eLife.

We agree that this work is based on observatory and exploratory data, which is typical of clinical data, as in our case. However, such studies are not without merits. One advantage is that our findings are based on human *in vivo* data, and are thus directly relevant to the clinical setting.

We thank the reviewer for the recognition of our methodology. We agree some of the results (including processes and pathways) described and reported here may had been implicated previously by us and others. However, we argue that such consistence in turn confirms the strength and quality of our data in confirming known results and thus predicting novel outcomes. Also, previous findings were based on various other data types, or even different species and were loosely disconnected from literature to literature. In this study, through integrative data and analyses, we present a more coherent, unifying picture from the human proteome perspective. We are re-submitting this data, methodology, and findings, together with a user-friendly web interface, as a resource paper, hoping that they would find use in the skeletal field, with potential wider audience in the proteomics community.